# Understanding Cross-layer Contributions to Mixture-of-Experts Routing in LLMs

**Wengang Li[1], Lingqi Zhang[2], Toshio Endo[1], Mohamed Wahib[2]**
[1]Institute of Science Tokyo    [2]RIKEN

## Abstract

Mixture-of-Experts (MoE) has been a prevalent method for scaling up large language models at a reduced computational cost. Despite its effectiveness, the routing mechanism of MoE still lacks a clear understanding from the perspective of cross-layer mechanistic interpretability. We propose a light-weight methodology at which we can break down the routing decision for MoE to the contribution of model components, in a recursive fashion. We use our methodology to dissect the routing mechanism by decomposing the input of routers into model components. We study how different model components contribute to the routing in different widely used open models. Our findings on four different LLMs reveal patterns such as: a) MoE layer outputs usually contribute more than attention layer outputs to the routing decisions of subsequent layers, b) *MoE entanglement* at which MoE firing up in layers consistently correlate with firing up of MoE in subsequent layers, and c) some components can persistently influence the routing in many following layers. Our study also includes findings on how different models have different patterns when it comes to long-range and short-range inhibiting/promoting effects that components can have over MoE in subsequent layers. Our results indicate the importance of quantifying the impact of components across different layers on MoE to understand the routing mechanism. [1]

## 1 Introduction

Transformer-based Large Language Models (LLMs) (Vaswani et al., 2017) have demonstrated powerful and versatile capabilities in recent years (Achiam et al., 2023; Comanici et al., 2025). To improve their capabilities, researchers have attempted to increase the model size as encouraged by scaling laws (Kaplan et al., 2020) and emergent abilities (Wei et al., 2022) of LLMs. However, this can lead to high computational cost. Mixture-of-Experts (MoE) (Jacobs et al., 1991; Shazeer et al., 2017; Fedus et al., 2022) has been applied in LLMs as an effective method to scale up models and alleviate those unfavorable effects, as it can reduce computation by routing the input to a subset of experts instead of using all model parameters to process it.

Although MoE have achieved great success in many advanced LLMs such as Grok-1 (xAI, 2024) and Gemini 2.5 (Comanici et al., 2025), there is still a lack of understanding of how components in different layers affect how the routing mechanism works, from the perspective of cross-layer mechanistic interpretability. Previous studies mainly investigated the routing mechanism at the expert-level. Muennighoff et al. (2025) and Jiang et al. (2024) performed analyses on the domain or token specialization of experts. Muennighoff et al. (2025) studied the co-activation of experts in the same layer. Lo et al. (2025) inspected the weights of MoE (including routers and experts), gate scores, and expert outputs using similarity and norm metrics. These studies heavily investigated the correlation between experts or between experts and tokens and overlooked the interaction between routers and other components in the model.[2]

In this work, we aim to explore and understand the routing mechanism by recursively decomposing the input of routers into components and studying how they contribute to the routing. In summary, our observations and conclusions, from studying four different models, are as follows:

---

[1]Code available at: https://github.com/wengangli/routing-contribution

[2]In this work, we mainly use the word "component" in a layer to refer to any of the following:, individual attention head, attention layer, individual expert in an MoE, and a full MoE FFN inclusive of all its experts.

• Our results reveal that MoE routing cannot be understood as a purely local process; rather, it emerges from intricate cross-layer interactions among model components.

• MoE layer output usually has the strongest influence on the routing rather than other components except for the first few layers. However, tokens and attention layer output may have a stronger influence in a small group area such as the bottom and the top layers.

• Routing is shaped not just by local computations, but also by long-range cross-layer entanglements, challenging the assumption that routing decisions are primarily local.

• A few experts have a significant influence on the routing decisions, consistent with the effect reported in (Su et al., 2025).

## 2  RELATED WORK

**Mixture of Experts.** Mixture-of-Experts (MoE) was first introduced in Jacobs et al. (1991) and has been studied for decades. Shazeer et al. (2017) proposed sparsely-gated MoE (SMoEs) as a method to scale up deep learning models with reduced inference overheads. They introduced the top-K routing algorithm, which has become a dominant paradigm nowadays for its simplicity and effectiveness. Researchers have put much effort in optimizing the design of SMoEs (Fedus et al., 2022). Recent studies have also attempted to understand the mechanism of the MoE layer (Chen et al., 2022) and have discovered the routing scores can be applied in model compression (Li et al., 2024) or used as an embedding model (Li & Zhou, 2025). Lo et al. (2025) made an early attempt to analyze MoE-based language models by observing the correlations and norms of some components related to the MoE layer, such as experts and gate scores.

**Decomposition of Transformers.** The output of Transformer blocks can be decomposed as a linear combination of outputs of its internal components, facilitating the dissection and understanding of Transformer-based language models (Elhage et al., 2021; Geva et al., 2021; Yu & Ananiadou, 2024; Ferrando & Voita, 2024). These methods decompose the outputs of attention or Feed-Forward Network (FFN) layers into smaller component vectors that can be further studied.The assignment scores assigned to experts can also be decomposed as the sum of sub-scores assigned by the components. Hence, we can study the distribution of the sub-scores to understand how these components influence routing decisions.

## 3  BACKGROUND

In this section, we present a recursive decomposition of the architecture of the MoE-based Transformer into components that together comprise the assignment score of MoE routing.

**MoE-based Transformer.** An MoE-based decoder-only Transformer consists of $L$ blocks. Each block consists of an attention layer, followed by a Mixture-of-Experts (MoE) layer. Given an input token embedding sequence $\boldsymbol{T} = [\boldsymbol{t}_1, \boldsymbol{t}_2, ..., \boldsymbol{t}_u]$ to the model, the first layer input $\boldsymbol{x}_{in,i}^0 \in \mathbb{R}^{d_e}$ is the token embedding $\boldsymbol{t}_i$, where $d_e$ is embedding dimension. The block output $\boldsymbol{x}_{out,i}^{\ell}$ (Token $i$, Block $\ell$) is formulated as follows:

$$\boldsymbol{x}_{out,i}^{\ell} = \boldsymbol{x}_{in,i}^{\ell} + \boldsymbol{a}_{out,i}^{\ell} + \boldsymbol{m}_{out,i}^{\ell}, \tag{1}$$

where $\boldsymbol{x}_{in,i}^{\ell}$ is the input of Block $\ell$ ($\boldsymbol{x}_{in,i}^{\ell} := \boldsymbol{x}_{out,i}^{\ell-1}$ for $\ell > 0$), $\boldsymbol{a}_{out,i}^{\ell}$ and $\boldsymbol{m}_{out,i}^{\ell}$ are the outputs of attention and MoE Layer $\ell$, respectively. The final block output $\boldsymbol{x}_{out,i}^{L-1}$ is normalized by layer normalization and then projected onto the vocabulary space to yield the probability distribution of the next token.

**Attention Layer.**[3] The attention layer output $\boldsymbol{a}_{out,i}^{\ell} \in \mathbb{R}^{d_e}$ can be decomposed into the linear combination of the head outputs $\boldsymbol{a}_{out,i}^{\ell,h}$, which can be further decomposed at the token level:

$$\boldsymbol{a}_{out,i}^{\ell} = \sum_{h=1}^{H} \boldsymbol{a}_{out,i}^{\ell,h} = \sum_{h=1}^{H} \sum_{p=1}^{i} \boldsymbol{W}_O^{\ell,h} \boldsymbol{A}_{i,p}^{\ell,h} \boldsymbol{v}_p^{\ell,h}. \tag{2}$$

---

[3]For simplicity, we focus on standard multi-head attention here.

The element of attention map $\boldsymbol{A}^{\ell,h} \in \mathbb{R}^{u \times u}$ is computed by:

$$\boldsymbol{A}_{i,p}^{\ell,h} = \begin{cases} \underset{1 \leq p \leq i}{\mathrm{softmax}}(\frac{\boldsymbol{q}_i^{\ell,h} \cdot \boldsymbol{k}_p^{\ell,h}}{\sqrt{d_h}}) & p \leq i \\ 0 & p > i \end{cases}, \tag{3}$$

where $d_h$ is head dimension, $\boldsymbol{q}_i^{\ell,h}, \boldsymbol{k}_p^{\ell,h}, \boldsymbol{v}_p^{\ell,h} \in \mathbb{R}^{d_h}$ are query, key, value vectors, respectively. Mathematically, $\boldsymbol{q}_i^{\ell,h} = \boldsymbol{W}_Q^{\ell,h} \boldsymbol{a}_{in,i}^{\ell}$, $\boldsymbol{k}_p^{\ell,h} = \boldsymbol{W}_K^{\ell,h} \boldsymbol{a}_{in,p}^{\ell}$, $\boldsymbol{v}_p^{\ell,h} = \boldsymbol{W}_V^{\ell,h} \boldsymbol{a}_{in,p}^{\ell}$, where $\boldsymbol{W}_Q^{\ell,h}, \boldsymbol{W}_K^{\ell,h}, \boldsymbol{W}_V^{\ell,h} \in \mathbb{R}^{d_h \times d_e}$, $\boldsymbol{W}_O^{\ell,h} \in \mathbb{R}^{d_e \times d_h}$ are query, key, value and output weight matrices of Head $h$ in attention Layer $\ell$, $d_h$ is head dimension, $\boldsymbol{a}_{in,i}^{\ell}$ is the attention layer input:

$$\boldsymbol{a}_{in,i}^{\ell} = \mathrm{LN}_i^{\ell}(\boldsymbol{x}_{in,i}^{\ell}), \tag{4}$$

where $\mathrm{LN}_i^{\ell}(\cdot)$ is layer normalization. RMS layer normalization is applied in the tested models in this work, hence $\mathrm{LN}_i^{\ell}(\boldsymbol{z}) := \frac{\boldsymbol{z} \cdot \gamma^{\ell}}{\mathrm{RMS}(\boldsymbol{z})}$, where $\boldsymbol{z} \in \mathbb{R}^{d_e}$ is the input vector, $\mathrm{RMS}(\cdot)$ is the root mean square function, and $\gamma^{\ell} \in \mathbb{R}^{d_e}$ is a learnable parameter.

**MoE Layer.** The MoE layer consists of a router and $N$ experts, i.e., $N$ parallel sub-FFN layers. The router assigns a **score** to each expert and selects the top-K experts to process the MoE layer input $\boldsymbol{m}_{in,i}^{\ell}$, where K is a hyperparameter. Mathematically,

$$\boldsymbol{m}_{in,i}^{\ell} = \mathrm{LN}_i^{\ell}(\boldsymbol{x}_{in,i}^{\ell} + \boldsymbol{a}_{out,i}^{\ell}). \tag{5}$$

Typically, the scores of all $N$ experts are computed by $\boldsymbol{W}_G^{\ell} \boldsymbol{m}_{in,i}^{\ell}$, where $\boldsymbol{W}_G^{\ell} \in \mathbb{R}^{N \times d_e}$ is the routing weight matrix. Each row vector $\boldsymbol{g} \in \mathbb{R}^{d_e}$ of the routing weight matrix $\boldsymbol{W}_G$ corresponds to one expert. We call these row vectors "**routing weight vectors**". The assignment score $S$ of Expert $(\ell, n)$, i.e., Expert $n$ in MoE Layer $\ell$, is essentially the dot product of its corresponding routing weight vector $\boldsymbol{g}^{\ell,n}$ and the MoE layer input $\boldsymbol{m}_{in,i}^{\ell}$:

$$S(\boldsymbol{g}^{\ell,n}, \boldsymbol{m}_{in,i}^{\ell}) = \boldsymbol{g}^{\ell,n} \cdot \boldsymbol{m}_{in,i}^{\ell}. \tag{6}$$

The assignment scores are passed through a Softmax function to yield expert weights. The MoE layer output $\boldsymbol{m}_{out,i}^{\ell}$ is the weighted sum of outputs of selected experts:

$$\boldsymbol{m}_{out,i}^{\ell} = \sum_{j \in J} r^{\ell,j}(\boldsymbol{m}_{in,i}^{\ell}) \boldsymbol{e}_{out,i}^{\ell,j}, \tag{7}$$

where $r^{\ell,j}(\cdot) = \underset{j \in J}{\mathrm{softmax}}(S(\boldsymbol{g}^{\ell,j}, \cdot))$ is the expert weight, $\boldsymbol{e}_{out,i}^{\ell,j}$ is the expert output, and $J$ is the set of indices of selected experts.

# 4 METHODOLOGY

For each given input token, the MoE router assigns scores to experts and selects the top-K experts to process the input. Assignment scores have two determinants: the routing weight vectors and the MoE layer inputs (Equation 6). Intuitively, assignment scores are approximately decomposable since the MoE layer input can be decomposed into components (Equations 1 and 5). To understand the routing mechanism, we can study the patterns of scores "assigned" by the components.[4] In this section, we delineate the decomposition method in Section 4.1 and discuss some basics of scoring patterns in Section 4.2.

## 4.1 DECOMPOSITION OF EXPERTS ASSIGNMENT SCORES

In this section, we discuss how the assignment score of an expert can be decomposed into sub-scores assigned by components of different granularities, from entire layers to individual neurons. We define an approximation operation $\overline{\mathrm{LN}}_i^{\ell}(\boldsymbol{c}) = \frac{\boldsymbol{c} \cdot \gamma^{\ell}}{\mathrm{RMS}(\boldsymbol{z})}$, where $\boldsymbol{c}$ is an arbitrary component and $\boldsymbol{z}$ is

---

[4]When we say the score assigned by a component to an expert, we refer to the portion of the expert's score contributed by the component.

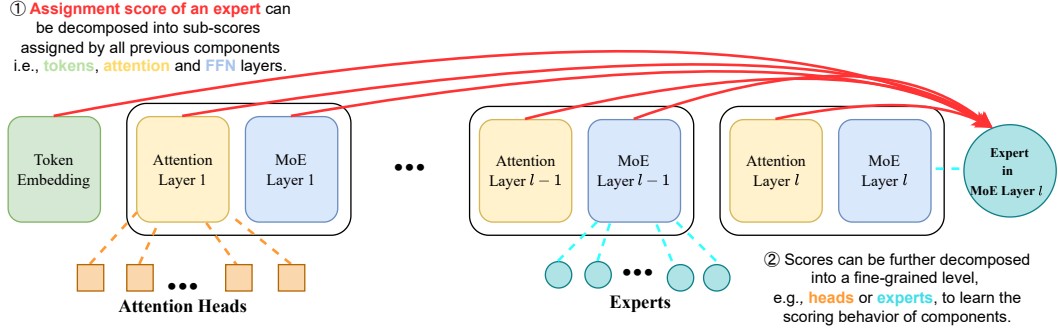

Figure 1: Overview of the decomposition of experts assignment scores.

the original input of $\text{LN}_i^\ell(z)$. This approximation distributes the same normalization factor across all components. We suggest that this is an acceptable definition for the contribution of $c$ to $\overline{\text{LN}}_i^\ell(c)$ in the context of studying routing decisions since it preserves the direction of the projected component and keeps their relative magnitudes faithful to what the router actually receives. We can recursively apply Equations 1 and 5 to decompose the score assigned to the Expert $(\ell, n)$:

$$
\begin{aligned}
S(\boldsymbol{g}^{\ell,n}, \boldsymbol{m}_{in,i}^\ell) &= \boldsymbol{g}^{\ell,n} \cdot \boldsymbol{m}_{in,i}^\ell = \boldsymbol{g}^{\ell,n} \cdot \text{LN}_i^\ell(\boldsymbol{x}_{in,i}^\ell + \boldsymbol{a}_{out,i}^\ell) \\
&= \boldsymbol{g}^{\ell,n} \cdot \text{LN}_i^\ell(\boldsymbol{x}_{in,i}^0 + \sum_{c=1}^\ell \boldsymbol{a}_{out,i}^c + \sum_{c=1}^{\ell-1} \boldsymbol{m}_{out,i}^c) \\
&= \boldsymbol{g}^{\ell,n} \cdot \overline{\text{LN}}_i^\ell(\boldsymbol{x}_{in,i}^0) + \boldsymbol{g}^{\ell,n} \cdot \sum_{c=1}^\ell \overline{\text{LN}}_i^\ell(\boldsymbol{a}_{out,i}^c) + \boldsymbol{g}^{\ell,n} \cdot \sum_{c=1}^{\ell-1} \overline{\text{LN}}_i^\ell(\boldsymbol{m}_{out,i}^c)
\end{aligned}
\tag{8}
$$

Equation 8 indicates that the assignment score can be decomposed into multiple sub-scores, i.e., the scores assigned by the token embedding ($S(\boldsymbol{g}^{\ell,n}, \boldsymbol{x}_{in,i}^0) = \boldsymbol{g}^{\ell,n} \cdot \overline{\text{LN}}_i^\ell(\boldsymbol{x}_{in,i}^0)$), attention layer outputs, and MoE layer outputs. Both attention and MoE layer outputs can be further decomposed. According to Equation 2, the score $S(\boldsymbol{g}^{\ell,n}, \boldsymbol{a}_{out,i}^c)$ assigned by attention layer output $\boldsymbol{a}_i^c$ can be further decomposed as follows:

$$
S(\boldsymbol{g}^{\ell,n}, \boldsymbol{a}_{out,i}^c) = \boldsymbol{g}^{\ell,n} \cdot \overline{\text{LN}}_i^\ell(\boldsymbol{a}_{out,i}^c) = \boldsymbol{g}^{\ell,n} \cdot \sum_{h=1}^H \overline{\text{LN}}_i^\ell(\boldsymbol{a}_{out,i}^{c,h}) = \boldsymbol{g}^{\ell,n} \cdot \sum_{h=1}^H \sum_{p=1}^i \overline{\text{LN}}_i^\ell(\boldsymbol{W}_O^{c,h} \boldsymbol{A}_{i,p}^{c,h} \boldsymbol{v}_p^{c,h}) \tag{9}
$$

From this perspective, we can regard the score assigned by an attention layer output to an expert as the sum of the scores assigned by tuples like (head, query, key) (Equations 3 and 9). Similarly, we can decompose the scores $S(\boldsymbol{g}^{\ell,n}, \boldsymbol{m}_{out,i}^c)$ assigned by the MoE layer output $\boldsymbol{m}_i^c$ into the scores assigned by the selected experts (Equation 7):

$$
S(\boldsymbol{g}^{\ell,n}, \boldsymbol{m}_{out,i}^c) = \boldsymbol{g}^{\ell,n} \cdot \overline{\text{LN}}_i^\ell(\boldsymbol{m}_{out,i}^c) = \boldsymbol{g}^{\ell,n} \cdot \sum_{j \in J} \overline{\text{LN}}_i^\ell(r^{c,j}(\boldsymbol{m}_{in,i}^c) \boldsymbol{e}_{out,i}^{c,j}) \tag{10}
$$

It is possible to further decompose the expert (i.e., sub-FFN) outputs at the neuron level (Geva et al., 2021; Dai et al., 2022; Geva et al., 2022):

$$
\boldsymbol{e}_{out,i}^{\ell,j} = \sum_{z=1}^{d_{\text{ffn}}} \boldsymbol{W}_{d_{(:,z)}}^{\ell,j} \cdot [\sigma(\boldsymbol{W}_{g_{(z,:)}}^{\ell,j} \cdot \boldsymbol{m}_{in,i}^\ell) \cdot (\boldsymbol{W}_{u_{(z,:)}}^{\ell,j} \cdot \boldsymbol{m}_{in,i}^\ell)], \tag{11}
$$

where all of the matrices correspond to Expert $(\ell, j)$, $\boldsymbol{W}_{d_{(:,z)}}^{\ell,j}$ is the $z$-th column of down-projection matrix, $\boldsymbol{W}_{g_{(z,:)}}^{\ell,j}$ and $\boldsymbol{W}_{u_{(z,:)}}^{\ell,j}$ are the $z$-th row of gating matrix and up-projection matrix, respectively. $d_{\text{ffn}}$ is denotes the intermediate dimension of FFN. $\sigma(\cdot)$ is an activation function. We leave it for further study, considering its potential complexity.

## 4.2 BASICS OF SCORING

In Section 4.1, we proposed a method for determining the scores of experts assigned by the components. In this section, we discuss what can be learned from the scores assigned by components. The proofs of our propositions are in Appendix A.

**Proposition 1: Variance of scores assigned by a component measures its influence on the routing decision.** Suppose a component assigns a constant score to all experts, which means its scoring variance is 0, then it does not influence the routing decisions because if the scores it assigns are dropped, the differences between the scores of experts are unchanged. Intuitively, we posit that a component with higher scoring variance has a stronger influence on the routing decisions and thus is more important. Based on this postulate, we can further infer that the length (i.e., L2-norm) of the component influences the upper bounds of the variance of scores assigned by it, which indicates that strong influences on the routing decisions tend to be associated with components with a large norm.

**Proposition 2: Positive scores promote experts, negative scores inhibit experts. The degree is measured by the score magnitude.** Since the scoring operation is a dot product of a gating weight vector and a component vector (Equations 6, 8, 9, and 10), the angle between them controls the sign of the score. When the angle is acute (obtuse), the score assigned by the component to the corresponding expert is positive (negative), indicating the component promotes (inhibits) the expert to be selected. When the two vectors are orthogonal, the score is zero, indicating the component has no bias on the corresponding expert. The magnitude of the score is controlled by the length of the two vectors and the angle between them. Hence, if we fix a gating weight vector, then a component with a smaller angle and larger length will assign a higher score to the corresponding expert.

## 5 SCORE DISTRIBUTION

In this section, we show empirical results on the assignment score distribution of experts assigned by tokens, attention layer outputs, and MoE layer outputs, respectively.

### 5.1 EXPERIMENTAL SETUP

**Models and dataset.** We adopt four MoE-based language models, scaling from OLMoE (Muennighoff et al., 2025), DeepSeek-V2-Lite (Liu et al., 2024), Qwen3-30B-A3B (Yang et al., 2025), to Mixtral-8x7B (Jiang et al., 2024). Their basic information is summarized in Appendix B. We randomly select samples with at least 32 tokens from C4 dataset (Raffel et al., 2020) and truncate each to the first 32 tokens to simplify the experiments. We use 1000 and 5000 samples for the experiments in Section 5.1 and 5.3, respectively. In this section, we report the results from OLMoE in the main text. The results of the other three models are reported in Appendix D.

**Metrics.** We use the variance of scores assigned by a component $c$ to the experts in an MoE layer to measure the contribution of the component to the routing decisions of that layer: The variance of scores $s_1, s_2, ..., s_N$ is $\frac{1}{N}\sum_{n=1}^{N}(s_n - \mu)^2$, where $\mu = \frac{1}{N}\sum_{n=1}^{N}s_n$. To find the scoring tendency (promotion or inhibition) of a component to a set of specified experts, we use the average positive score (APS) and average negative score (ANS) assigned to those experts by the component:

$$APS = \frac{1}{N}\sum_{n=1}^{N}S(\boldsymbol{g}_j, \boldsymbol{c})\mathbb{1}_{S(\boldsymbol{g}_j, \boldsymbol{c})>0}, \quad ANS = \frac{1}{N}\sum_{n=1}^{N}S(\boldsymbol{g}_j, \boldsymbol{c})\mathbb{1}_{S(\boldsymbol{g}_j, \boldsymbol{c})<0}, \tag{12}$$

where $N$ is the number of experts and $\mathbb{1}$ is the indicator function. We separate the positive and negative scores to avoid cancellation. Variance, APS, and ANS are computed per token position, then averaged over non-lead tokens, and then averaged over samples. We report the results related to the lead position in Appendix D since we observe that they have a different behavior, compared with tokens on other positions.

### 5.2 RESULTS ON TOKEN SCORING

**Token scoring is influenced by its part-of-speech (POS).** We pack the scores of all experts assigned by each token into a vector and apply t-SNE(Maaten & Hinton, 2008) to visualize them. [5] From Figure 2a, we find that most scores assigned by tokens are clustered according to the POS tags of tokens, which implies that the POS influences token scoring. Furthermore, most function words are clustered into isolated lumps. In contrast, content words are entangled, which is possibly because

---

[5]If a word is split into multiple tokens, each token inherits the POS of the word.

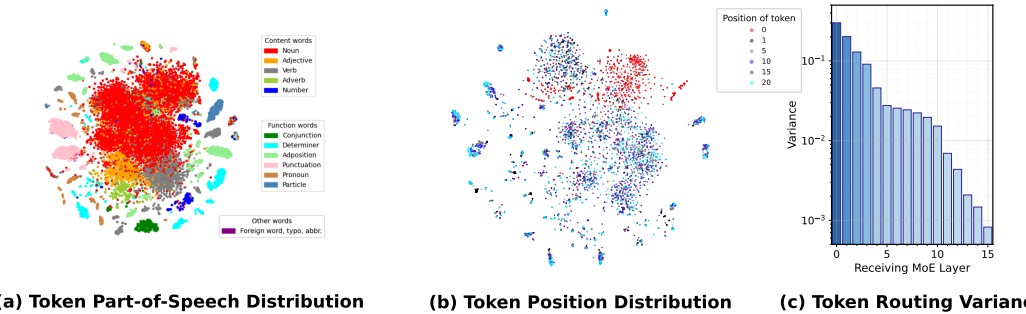

(a) Token Part-of-Speech Distribution    (b) Token Position Distribution    (c) Token Routing Variance

Figure 2: (a) t-SNE of scores assigned by token embeddings, colored by POS. (b) t-SNE of scores assigned by token embeddings, colored by position. (c) Average variances of scores assigned by tokens to MoE layers.

function words may have multiple POS, e.g., "drink" can be a noun or a verb, whereas the semantics of function words is more stable.

**The lead token is special.** In Figure 2b, the lead tokens of each prompt appear clustered, whereas other tokens, in comparison, are not, indicating that the lead position has a noticeable influence on scoring. We find the scoring distribution of attention and MoE layers at the lead position also has a typical pattern. We speculate that the attention layers capture the position information, lead to the norm of MoE layer output at the lead position to being typically larger than that at other tokens, which influences the RMS term of layer normalization and finally influences the scoring distribution. In light of this, we report the results excluding the lead position in the main text henceforth and report the results related to the lead position in Appendix D.

In Figure 2c, the average variance of scores assigned by tokens decreases rapidly as the layer goes deeper, showing that the influence of tokens on routing decisions decreases rapidly. Furthermore, the variance of scores in the first two MoE layers assigned by tokens is typically high, indicating the tokens have a strong influence on the routing decisions in them, which is in line with intuition. The magnitude of ANS and APS also decreases as the layer goes deeper (see Appendix D).

## 5.3 RESULT ON ATTENTION/MoE LAYER OUTPUTS

In this section, we analyze the average scoring variance of the attention and MoE layer outputs and the average positive/negative scores assigned by them, to study their influence on the routing decisions.

**Average variance.** As shown in Figure 3a, the highest variance of attention layer scoring occurs at A0 → M0, i.e., the assignment from sending attention Layer 0 to receiving MoE Layer 0. The variance diminishes gradually as the receiving layer appears deeper. Comparatively, the first two attention layers (A0 and A1) exhibit a higher variance than the intermediate sending attention layers, although the magnitude of their scoring variance is still low. We compare the four models we tested and find that the early sending attention layers generally have a high average variance to their neighbor receiving layers. Surprisingly, a few MoE layers have a pronounced *entangled* influence on the routing decisions on following MoE layers (Figure 3d): Sending layers M1 and M4 have a notably higher variance compared with others, indicating they have a higher importance. It is also unique that their variance does not decrease monotonically. We will show that two experts with typically high variance in these two layers cause the "stripes" (Section 7). We also find such stripes in DeepSeek, Qwen and Mixtral, but some of them may just have an apparently higher variance compared with neighbor sending layers, instead of having an increasing average variance on the receiving layers. The stripes also occur in sending attention layers in Qwen. Finally, the variance of scores assigned by sending MoE layers is usually comparatively higher than that assigned by sending attention layers, except for the bottom receiving layers M0~M3, indicating that the MoE layer output usually has a stronger influence on the routing decisions than the attention layer output.

These findings suggest two key implications. First, load balancing of expert parallelism can be improved by prefetching and preloading experts in high-variance layers, leveraging cross-layer entanglement to reduce contention. Second, post-NAS approximation strategies Gu et al. (2025) can

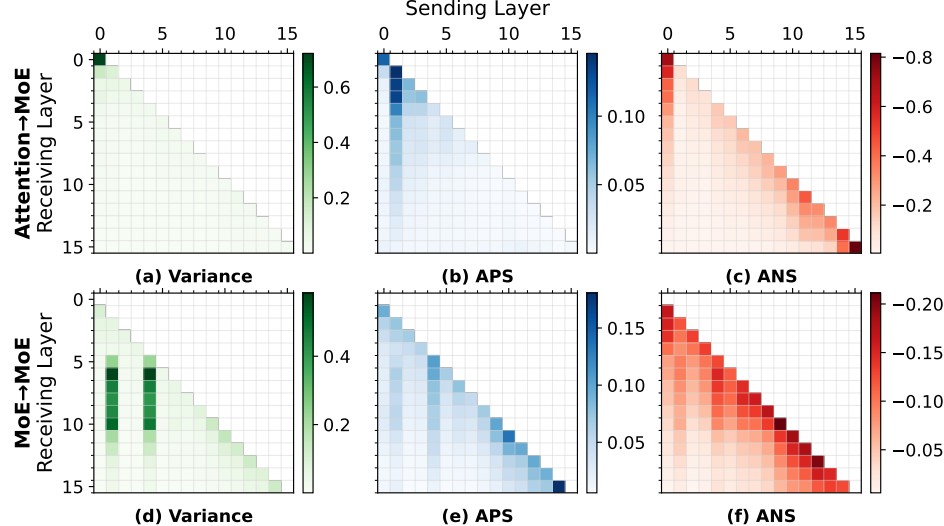

Figure 3: Scores assigned by attention Layer x to MoE Layer y: (a) Variance, (b) Average positive scores (APS), (c) Average negative scores (ANS); Scores assigned by MoE Layer x to MoE Layer y: (d) Variance, (e) APS, (f) ANS.

selectively compress low-variance attention layers while preserving influential ones, enabling more efficient yet accurate architectures.

**Scoring pattern of layers.** We conducted an experiment to analyze the scores assigned to all the experts, rather than the selected ones. By observing ANS and APS, we can learn which components promote or inhibit the experts in a specific layer. We first look at the scoring pattern of sending attention layers. Comparing Figure 3b and c, we note that the positive and negative scores occur in different areas: positive scores occur at the early sending layers i.e., left side of x-axis (A0 and A1), whereas negative scores occur at A0, and most areas on or near the diagonal, and the magnitude increases as the sending layer goes deeper. In other words, promotion effects—components assigning positive scores to experts—tend to be local, whereas inhibition effects are more global, with components exerting stronger inhibition as the receiving MoE layer appears deeper in the model. Comparing the tested models, we find that the negative scores tend to occur at the bottom or top sending attention layers. The magnitude of APS is generally smaller than that of ANS, although Mixtral seems to be an exception.

The scoring pattern of sending MoE layers is shown in Figure 3e and f. Sending M1 and M4 have relatively high positive and negative scores, respectively, which conforms to the variance matrix (Figure 3d). The positive scores mainly appear at sending M1, M4, deeper sending layers (M9 $\sim$ M14) and the diagonal, whereas the negative scores appear at sending M1, M4, and the area near the diagonal. We also find the APS and ANS matrices in Qwen have a multi-stripe pattern, i.e., more prominent entanglement effect, whereas other tested models show only a few or no stripes.

# 6 SCORING OF ATTENTION HEADS

In this section, we investigate the scores assigned by attention heads to the experts in different layers. We continue to use C4 dataset to observe the general behavior of the scoring. Additionally, we use Indirect Object Identification (IOI) (Wang et al., 2023) task to study how function heads influence the routing decisions and how the variance of scores relate to the scoring patterns.

## 6.1 EXPERIMENTAL SETUP

**General test.** We conduct the general test on OLMoE and DeepSeek (Appendix G), which have fewer heads (256 and 432) and thus facilitate our analysis. We follow the setting in Section 5.1 and use 5000 samples for experiments.

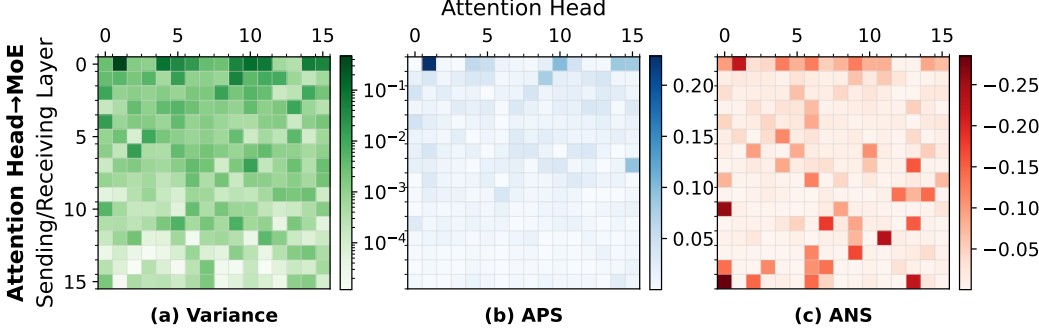

Figure 4: Scores assigned by heads to the experts in the same block: (a) Variance, (b) Average positive scores (APS), (c) Average negative scores (ANS).

**IOI task** is to predict the next token of a prompt like "*When Mary and John went to the store, John gave a drink to ___*", where the name that exists in the first clause but does not appear in the second clause is expected to be the prediction result, i.e., "Mary". We adopt this task to observe if function heads have a noticeable influence on the routing decisions. We use path patching (Wang et al., 2023) to find a portion of the function heads in OLMoE.[6] We can regard the scores assigned by a head as a "score map" once we determine a metric for tuples like (head, query, key). We use the variance of scores assigned by the tuple (head, query, key) to all experts in an MoE layer as the metric. We refer to it as the *score variance map* and compare it with the results from path patching and the attention map.

## 6.2 RESULTS ON GENERAL TEST

Since attention layer outputs usually exert the strongest influence on the routing decisions in the same block in OLMoE, we examine the variance and mean of scores assigned by heads to the experts in the same block to facilitate the observation (Figure 4). Henceforth, we use "AxHy" to denote Head y at attention Layer x. Heads with high variance mainly occur at the early layers (typically A0 and A1), conforming to the results in 3a. By comparing Figures 4b and c, we find that many attention heads tend to assign negative scores to the experts, and a small proportion of heads tend to generally assign positive scores, such as A7H15 and A9H6. The scores assigned by some heads are apparently polarized, such as A0H1. Finally, we observe that if a head has a relatively higher scoring variance, the absolute magnitude of scores assigned by the head is usually larger.

## 6.3 RESULTS ON IOI TASK

We employ IOI task as an example to study the scoring pattern of function heads. The visualized results are shown in Appendix F. Our main findings on this task are as follows:

**Function heads have a noticeable influence on the routing decisions.** We find that function heads, i.e., attention heads that contribute to finish the IOI task, usually have a higher scoring variance, compared with the heads do not show any clear function in the task, indicating that the function heads tend to have a stronger influence on the routing decisions than the other heads.

**Scores variance of attention heads correlate with attention maps.** We compare the "attention map" $\boldsymbol{A}_{i,p}^{x,y}$ (where query token $\boldsymbol{t}_i$ and key token $\boldsymbol{t}_p$ are fixed, Layer $x$ and Head $y$ are variables), with the "score variance map", that is, variance of scores assigned by tuples (head=AxHy, query=$\boldsymbol{t}_i$, key=$\boldsymbol{t}_p$), in other words, terms $\overline{\text{LN}}_i^c(\boldsymbol{W}_O^{x,y} \boldsymbol{A}_{i,p}^{x,y} \boldsymbol{v}_p^{x,y})$ for AxHy (Equation 9), to the experts in a specific layer $c(c \geq x)$. We find that they have a similar pattern, which is in line with our intuition since attention map can manipulate the attention output and thus influence the scoring.

---

[6]For simplicity, we do not aim to discover all the function heads activated in the IOI task in this work.

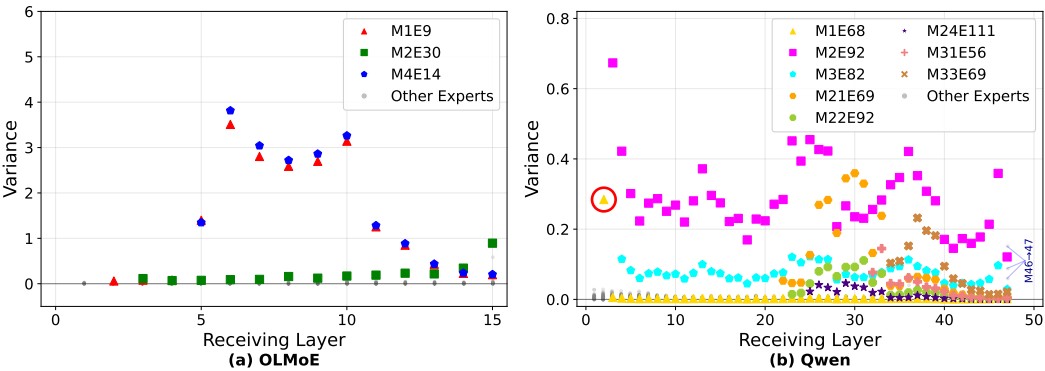

Figure 5: The variance of scores assigned by experts to the following layers.

# 7 SCORING OF EXPERTS

In Section 7.1, we use variance to measure the influence of experts on routing decisions in subsequent layers. In Section 5.3, we have seen that the variance of scores assigned by M1 and M4 in OLMoE has an unusual phenomenon: the variance does not decrease monotonically. We find that two experts contribute to the phenomenon. Furthermore, a small subset of experts has a strong influence on the routing decisions in OLMoE and Qwen3. Intriguingly, among these experts, some maintain their influence on the routing decisions till the last layers rather than decreasing significantly. We further propose a new metric named "average absolute ranking variation of top-K experts (AARV)" and use it to conduct a small causal analysis to observe a large-variance expert found in Section 7.1 as a case study in Section 7.2.

## 7.1 MAIN FINDINGS

**OLMoE (Figure 5a).** M1E9 (Experts 9 at MoE Layer 1) and M4E14 exhibit an unusual phenomenon where their influence peaks around MoE Layer 6, followed by a secondary peak near MoE Layer 10, then consistently decreases (with M1E9 dropping below M4E14 by the end). In contrast, M2E30 shows a steadily increasing influence that reaches its peak in the final layer.

**Qwen (Figure 5b).** M1E68 (circled in red) exhibits a strong but localized influence primarily on the immediate next layer. M2E92 consistently maintains the highest impact across all subsequent layers. M3E82 also shows persistent influence throughout, though with smaller magnitude. Other highlighted experts (M21E69, M22E92, M24E111, M31E56, M33E69) from middle and late layers generally follow the typical pattern of gradually building to peak influence before fading out.

We find some of these experts are the "Super Experts" identified by Su et al. (2025), which have an output of extreme magnitude. We summarize the rank of variance of scores assigned by the Super Experts to the experts in the next MoE layers in Appendix H. We find that not all the Super Experts have typically large scoring variance, such as M2E54 in DeepSeek. Although M1E68 in Qwen has the rank 1 variance among the experts in Layer 1, but its scoring variance decreases drastically (Figure 5b). The Super Experts in DeepSeek do not have a top rank nor a persistent high scoring variance. The scoring variance of Super Expert in Mixtral (M1H3) has a unique distribution: it has a relatively high variance in the layer $28 \sim 31$ (Appendix H).

## 7.2 CAUSAL ANALYSIS OF M1E9 IN OLMoE

In this section, we analyze M1E9 to study how it perturbs the top-K experts with a physical metric - "average absolute ranking variation of top-K experts (AARV)" in an MoE layer as a case study to directly justify that variance can influence the selection of top-K experts. We define the metric as follows:

$$AARV = \frac{1}{K} \sum_{e \in \text{top-}K} |\text{rank}_{\text{orig}}(e) - \text{rank}_{\text{pert}}(e)|, \tag{13}$$

where $e$ are top-K experts in an MoE layer, $\text{rank}_{\text{orig}}(e)$ is the original rank before any perturbations on the net assignment scores of Expert $e$, $\text{rank}_{\text{pert}}(e)$ is the perturbed rank after we set the score assigned by a component to zero.

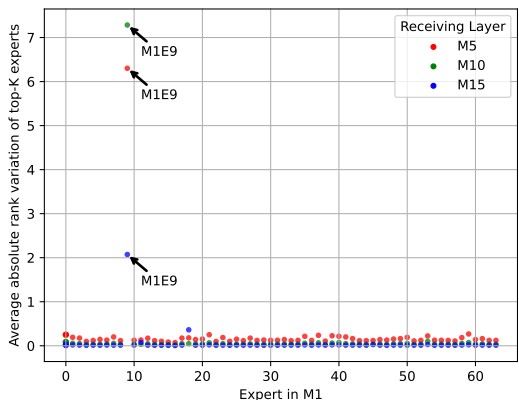

Figure 6: The AARV of M5, M10, and M15 when the scores assigned by an expert in M1 is set to zero.

We set the scores assigned by each expert in M1 to zero and show AARV of top-K experts in M5, M10, and M15 in Figure 6. We observe that M1E9 has a dominant influence on the routing decisions in these three layers, whereas the other experts in M1 weakly nudge the top-K decisions. The Spearman's coefficient results of the data in Figure 6 is provided in Appendix N.

We further conduct an ablation experiment and identify that M1E9 controls the activation of M4E14 and they need to coexist to exert a prominent influence on the routing decisions in MoE Layer 5 and the following layers (Appendix I).

## 8  SUMMARY OF FINDINGS AND CONCLUSION

In this work, we proposed a recursive decomposition framework to quantify how different components contribute to routing decisions in Mixture-of-Experts (MoE) language models. By breaking down expert assignment scores into contributions from tokens, attention layers, MoE outputs, and attention heads, we provided the first cross-layer perspective on routing interpretability. Our analysis across four production MoE models (OLMoE, DeepSeek-V2-Lite, Qwen3-30B-A3B, and Mixtral-8x7B) revealed several consistent patterns. **First**, MoE outputs exert the strongest and most persistent influence on downstream routing, while attention layers and tokens have more localized effects, especially in the bottom and top layers. **Second**, routing decisions are shaped by both promotion and inhibition: positive contributions typically act locally, while negative contributions inhibit experts across longer ranges. **Third**, we identified cross-layer entanglement phenomena, where certain MoE layers (e.g., M1, M4 in OLMoE) disproportionately affect routing in much deeper layers, forming "stripes" of influence. At a finer granularity, we found that a small set of experts and attention heads dominate the routing behavior, with some maintaining their impact throughout the model. Notably, not all previously identified "Super Experts" exhibit strong influence under our variance-based analysis. These findings demonstrate that MoE routing is not solely a local mechanism, but instead emerges from a complex interplay of components across layers. Our work shows that the routing decisions can be studied in a decomposition perspective to understand its underlying mechanism.

## 9  USE OF LARGE LANGUAGE MODELS (LLMs)

We do not use LLM in idea formation, experimental result analysis, or paper writing in this work. The authors take full responsibility for the contents in this paper.

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

# APPENDIX

## A  PROOF OF PROPOSITIONS

**Variance of scores assigned by a component measures its influence on the routing decision.**

**Proposition 1 (Variance of component-contributed scores and norm of component).**  Fix a layer $\ell$ and position $i$. Let $\boldsymbol{z}_c \in \mathbb{R}^{d_e}$ denote the contribution of a single component (e.g., token embedding, an attention head output, or a previous MoE block output) to the MoE-input before normalization. Define

$$\overline{\mathrm{LN}}_i^\ell(\boldsymbol{z}_c) \;=\; \frac{\gamma^\ell \odot \boldsymbol{z}_c}{\mathrm{RMS}\big(\boldsymbol{x}_{in,i}^\ell + \boldsymbol{a}_{out,i}^\ell\big)},$$

where $\boldsymbol{A} \in \mathbb{R}^{d_e \times d_e}$ is a fixed linear map for this $(\ell, i)$. Let $G \in \mathbb{R}^{N \times d_e}$ stack the routing weight vectors as rows, $G = \begin{bmatrix} (\boldsymbol{g}^{\ell,1})^\top \\ \cdots \\ (\boldsymbol{g}^{\ell,N})^\top \end{bmatrix}$. The vector of expert scores contributed by this component is then

$$\boldsymbol{s}^{(c)} \;=\; G\,\overline{\mathrm{LN}}_i^\ell(\boldsymbol{z}_c) \;=\; G\boldsymbol{A}\,\boldsymbol{z}_c \;\in\; \mathbb{R}^N, \quad \text{i.e.,} \quad s_n^{(c)} \;=\; (\boldsymbol{A}^\top \boldsymbol{g}^{\ell,n})^\top \boldsymbol{z}_c.$$

*(i) Zero-variance implies no routing influence.* If $\boldsymbol{s}^{(c)}$ is constant across experts, i.e., $s_1^{(c)} = \cdots = s_N^{(c)} = c$, then adding or removing this component shifts all experts by the same constant. For any pair $n, m$,

$$(s_n^{\mathrm{total}} + c) - (s_m^{\mathrm{total}} + c) = s_n^{\mathrm{total}} - s_m^{\mathrm{total}},$$

so the ordering is unchanged. Hence a constant-score component has variance 0 and no influence on routing.

*(ii) Variance as an influence measure.* Define $\mathrm{Var}(\boldsymbol{s}^{(c)}) := \frac{1}{N}\sum_{n=1}^N (s_n^{(c)} - \overline{s}^{(c)})^2$, where $\overline{s}^{(c)} = \frac{1}{N}\sum_n s_n^{(c)}$. Then $\mathrm{Var}(\boldsymbol{s}^{(c)}) = 0$ iff $\boldsymbol{s}^{(c)}$ is constant. Moreover, for any $\alpha \in \mathbb{R}$,

$$\mathrm{Var}(G\boldsymbol{A}(\alpha\boldsymbol{z}_c)) = \alpha^2\,\mathrm{Var}(G\boldsymbol{A}\boldsymbol{z}_c),$$

so larger component magnitudes yield quadratically larger variance, hence stronger influence on routing.

*(iii) Norm-controlled bounds.* By Cauchy–Schwarz,

$$|s_n^{(c)}| \;=\; \big|(\boldsymbol{A}^\top \boldsymbol{g}^{\ell,n})^\top \boldsymbol{z}_c\big| \;\le\; \|\boldsymbol{A}^\top \boldsymbol{g}^{\ell,n}\|_2\,\|\boldsymbol{z}_c\|_2.$$

Let $M = \max_n \|\boldsymbol{A}^\top \boldsymbol{g}^{\ell,n}\|_2$. Then

$$s_n^{(c)} \in [-M\|\boldsymbol{z}_c\|_2,\, M\|\boldsymbol{z}_c\|_2] \quad \forall n,$$

so the range of component-contributed scores is bounded by $2M\|\boldsymbol{z}_c\|_2$.

Additionally,

$$\mathrm{Var}(\boldsymbol{s}^{(c)}) = \frac{1}{N}\sum_{n=1}^N (s_n^{(c)} - \overline{s}^{(c)})^2 \;\le\; \frac{1}{N}\sum_{n=1}^N (s_n^{(c)})^2 = \frac{1}{N}\|\boldsymbol{s}^{(c)}\|_2^2 = \frac{1}{N}\|G\boldsymbol{A}\boldsymbol{z}_c\|_2^2 \le \frac{1}{N}\|G\boldsymbol{A}\|_F^2\,\|\boldsymbol{z}_c\|_2^2.$$

Consequently,

$$\mathrm{Var}(\boldsymbol{s}^{(c)}) \;\le\; \frac{1}{N}\|G\boldsymbol{A}\|_F^2\,\|\boldsymbol{z}_c\|_2^2,$$

showing that variance (and hence influence) is upper-bounded by a constant—depending on the router and normalization—times the squared L2 norm of the component.

*Conclusion.* Components that assign constant scores exert no influence, while those with larger norms admit wider score ranges and potentially larger variance across experts, thereby possessing greater capacity to alter expert rankings and influence routing.

**Proposition 2 (Sign and magnitude of component-contributed scores).** Fix a layer $\ell$, position $i$, and an expert $n$. Let the component contribution before routing be $\boldsymbol{z}_c \in \mathbb{R}^{d_e}$ and let $\boldsymbol{u} := \overline{\mathrm{LN}}_i^\ell(\boldsymbol{z}_c) \in \mathbb{R}^{d_e}$ denote its normalized contribution at this $(\ell, i)$ (cf. Eq. 8). The score contributed by this component to expert $n$ is

$$s_n^{(c)} = \boldsymbol{g}^{\ell,n} \cdot \boldsymbol{u}.$$

Let $\theta_n \in [0, \pi]$ be the angle between $\boldsymbol{g}^{\ell,n}$ and $\boldsymbol{u}$.

*(i) Sign determines promotion vs. inhibition.* By the cosine formula for the dot product,

$$s_n^{(c)} = \|\boldsymbol{g}^{\ell,n}\|_2 \|\boldsymbol{u}\|_2 \cos\theta_n.$$

Hence $s_n^{(c)} > 0$ iff $\theta_n \in (0, \frac{\pi}{2})$ (acute), $s_n^{(c)} < 0$ iff $\theta_n \in (\frac{\pi}{2}, \pi)$ (obtuse), and $s_n^{(c)} = 0$ iff $\theta_n = \frac{\pi}{2}$ (orthogonal). Since (a) top-K selection depends only on score orderings and (b) the softmax used to form expert weights is strictly increasing in each coordinate, adding a component with $s_n^{(c)} > 0$ increases expert $n$'s total score and softmax weight (promotes selection), while $s_n^{(c)} < 0$ decreases them (inhibits selection); $s_n^{(c)} = 0$ leaves them unchanged.

*(ii) Magnitude quantifies degree of influence.* The absolute score satisfies

$$|s_n^{(c)}| = \|\boldsymbol{g}^{\ell,n}\|_2 \|\boldsymbol{u}\|_2 |\cos\theta_n| \leq \|\boldsymbol{g}^{\ell,n}\|_2 \|\boldsymbol{u}\|_2,$$

with equality iff $\boldsymbol{g}^{\ell,n}$ and $\boldsymbol{u}$ are colinear. For fixed $\boldsymbol{g}^{\ell,n}$, the dependence on angle and component length is monotone:

$$\frac{\partial s_n^{(c)}}{\partial \theta_n} = -\|\boldsymbol{g}^{\ell,n}\|_2 \|\boldsymbol{u}\|_2 \sin\theta_n \leq 0, \qquad \frac{\partial s_n^{(c)}}{\partial \|\boldsymbol{u}\|_2} = \|\boldsymbol{g}^{\ell,n}\|_2 \cos\theta_n.$$

Thus, for $\theta_n \in [0, \frac{\pi}{2})$, decreasing the angle (better alignment) or increasing the component norm strictly increases $s_n^{(c)}$; for $\theta_n \in (\frac{\pi}{2}, \pi]$, the same operations make $s_n^{(c)}$ more negative, strengthening inhibition. Consequently, the *degree* of promotion/inhibition is exactly captured by the magnitude $|s_n^{(c)}|$, which grows with both alignment (via $|\cos\theta_n|$) and component length $\|\boldsymbol{u}\|_2$ (and hence with $\|\boldsymbol{z}_c\|_2$ up to the fixed normalization factor at $(\ell, i)$).

*Conclusion.* The sign of the component-expert dot product governs whether the component promotes or inhibits that expert's selection, while its magnitude—$\|\boldsymbol{g}^{\ell,n}\|_2 \|\boldsymbol{u}\|_2 |\cos\theta_n|$—quantifies the strength of this effect. $\square$

## B   BASIC INFORMATION OF TESTED MODELS

Table 1: Basic information of tested models.

| Information | OLMoE | DeepSeek-V2-Lite | Qwen3-30B-A3B | Mixtral-8x7B |
|---|---|---|---|---|
| Total Params | 7B | 16B | 30B | 47B |
| Activated Params | 1B | 3B | 3B | 7B |
| Number of Layers | 16 | 27 | 48 | 32 |
| Number of Routed Experts | 64 | 64 | 128 | 8 |
| Top-K | 8 | 6 | 8 | 2 |

NOTE: DeepSeek-V2-Lite has an FFN layer instead of an MoE layer in Layer 0. Each MoE layer has two shared experts in DeepSeek-V2-Lite.

## C   BRIEF INTRODUCTION TO IOI TASK

We apply the method (path patching) and the metric (logit difference) from (Wang et al., 2023) to reproduce the experiment on OLMoE, and identify four types of heads which are active at END token, as follows:

- **Name Mover Heads** attend to the previous name tokens. They promote the name tokens.

- **Negative Name Mover Heads** are similar to Name Mover Heads but inhibit the name tokens as the prediction result.

- **S-Inhibition Heads** inhibit S token, and influence Name Mover Heads and Negative Name Mover Heads.

- **Backup Name Mover Heads** are active when the Name Mover Heads are ablated. They also show a weak influence when the regular Name Mover Heads work normally.

We adapt the code provided in the original paper to generate 5000 samples for the IOI task experiments.

## D SUPPLEMNTARY RESULTS ON THE DECOMPOSITION AT THE LAYER LEVEL

We find that the scoring distribution at the lead tokens is completely different from other tokens (Figures 7 ~ 10). For example, In OLMoE, in the score assignment from sending attention layers to receiving MoE layers, the high variance occurs at the bottom sending layers to their neighbor receiving layers, and the last receiving layer. However, when sending and receiving layers are both MoE layers, the high variance occurs at some sending MoE layers and the last receiving MoE layer. The highest APS occur at A15 → M15, and M14 → M15. The highest ANS occur at A14 → M15 (A0 → M0), and M2 → M15. The four models at the lead token have different variance patterns. We can observe that there are strides (e.g., Figure 9d) in the score variance distribution, indicating that some sending layers are more influential and have a persistent influence on the routing decisions. The results at other tokens on DeepSeek, Qwen, and Mixtral are presented in Figures 11~13, respectively. The results on the scoring of tokens are presented in Figure 14.

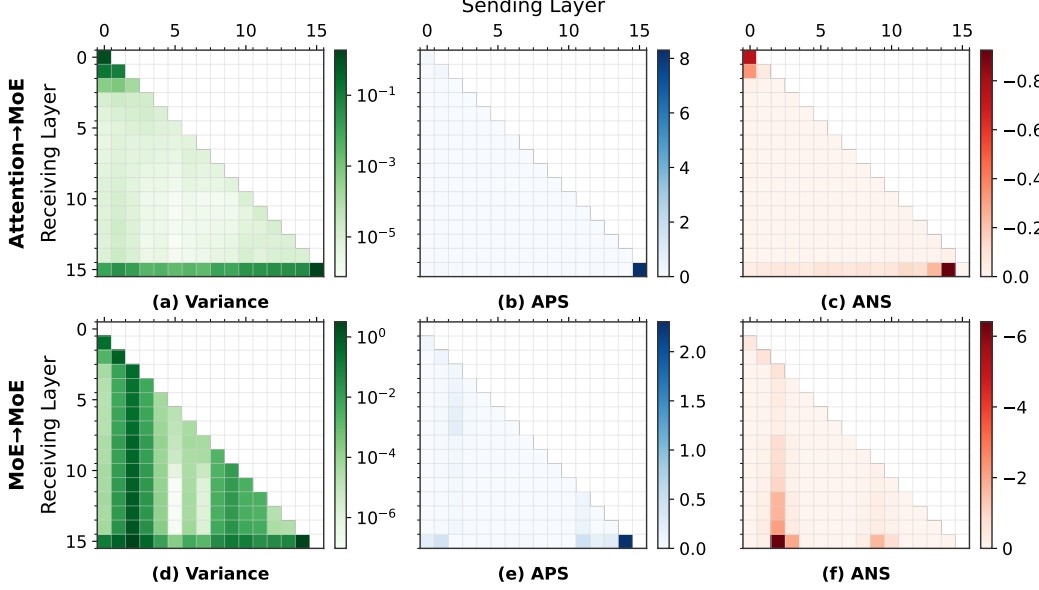

Figure 7: Scoring distribution at the lead tokens, OLMoE

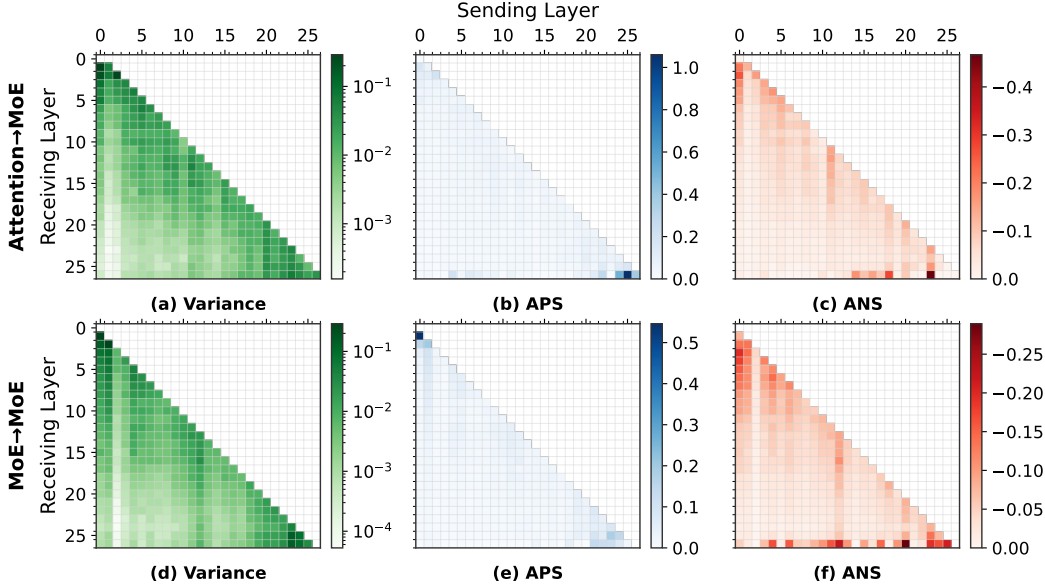

Figure 8: Scoring distribution at the lead tokens, DeepSeek

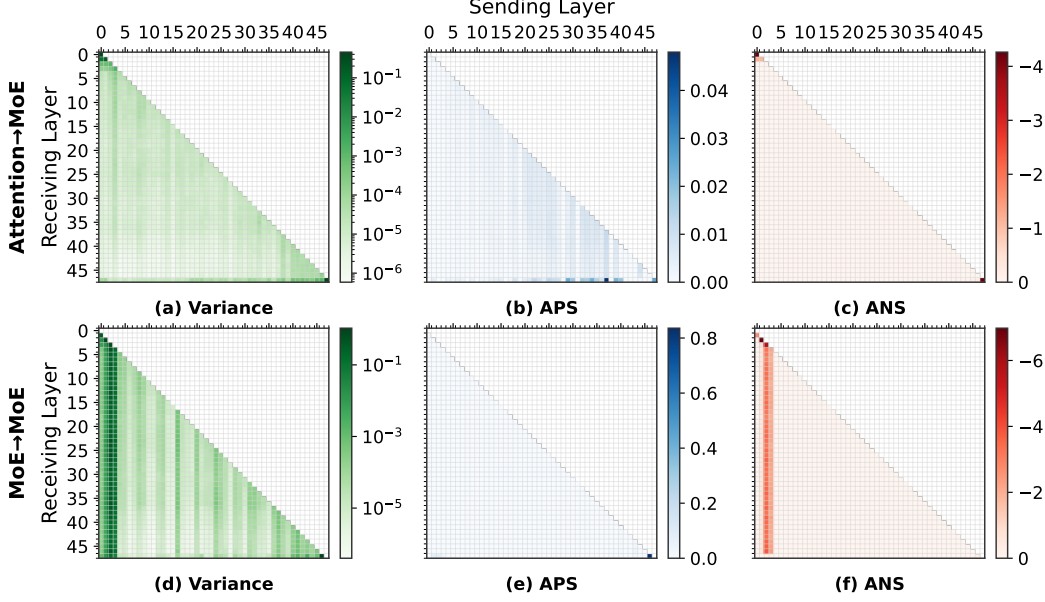

Figure 9: Scoring distribution at the lead tokens, Qwen

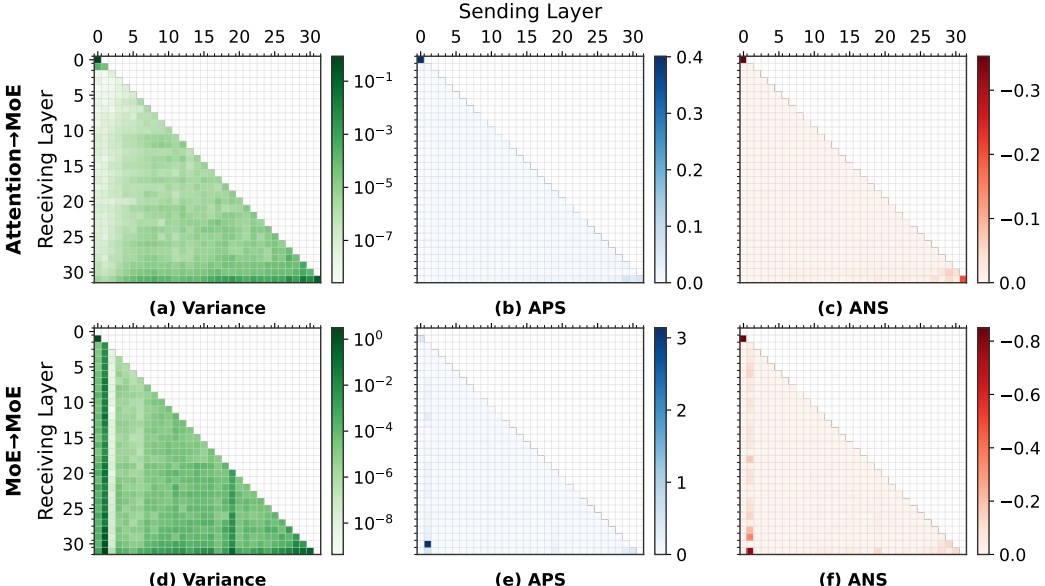

Figure 10: Scoring distribution at the lead tokens, Mixtral

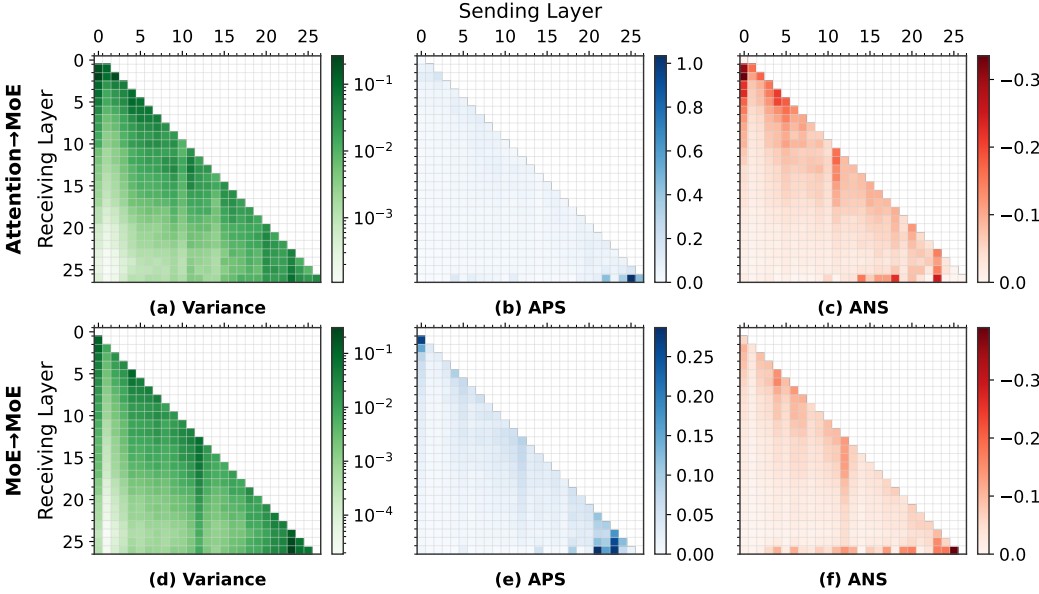

Figure 11: Scoring distribution at other tokens (except the lead tokens), DeepSeek

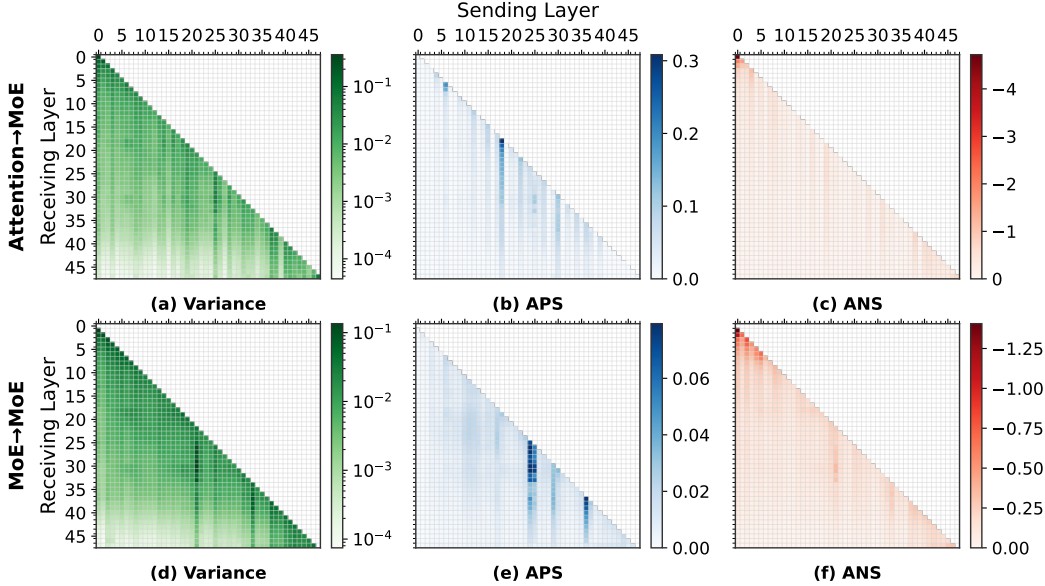

Figure 12: Scoring distribution at other tokens (except the lead tokens), Qwen

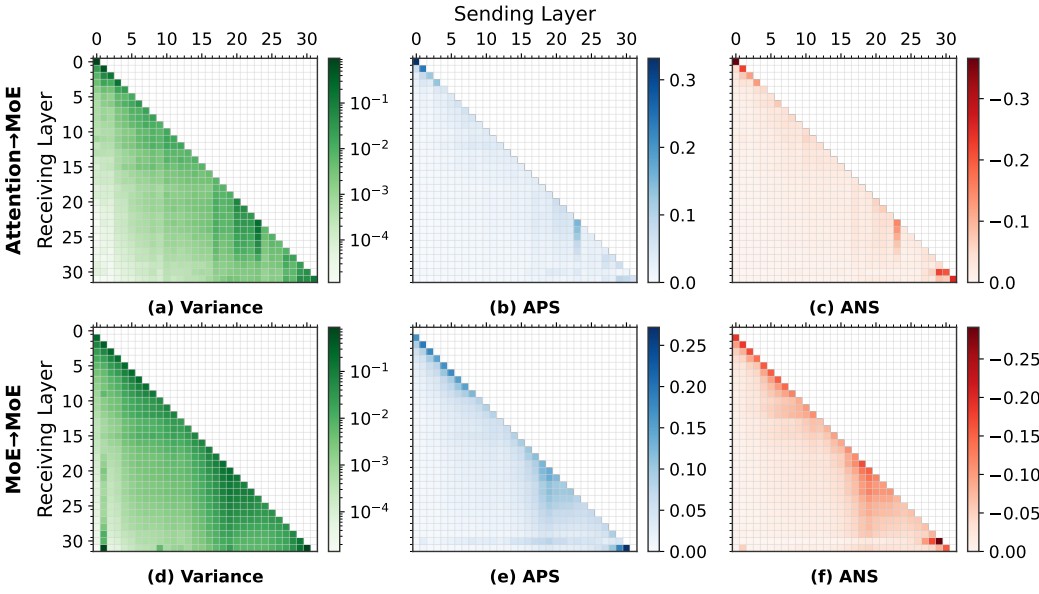

Figure 13: Scoring distribution at other tokens (except the lead tokens), Mixtral

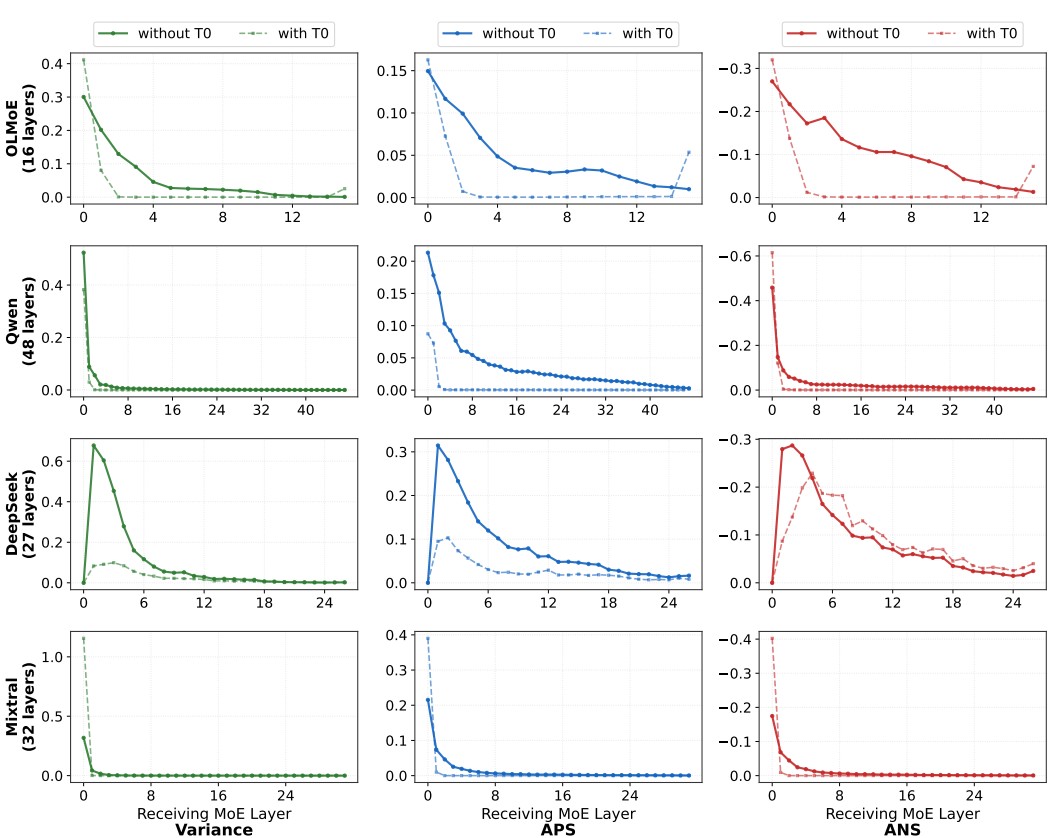

Figure 14: Scoring distribution of tokens.

# E  RESULTS OF T-SNE ON SCORES ASSIGNED BY TOKENS (QWEN)

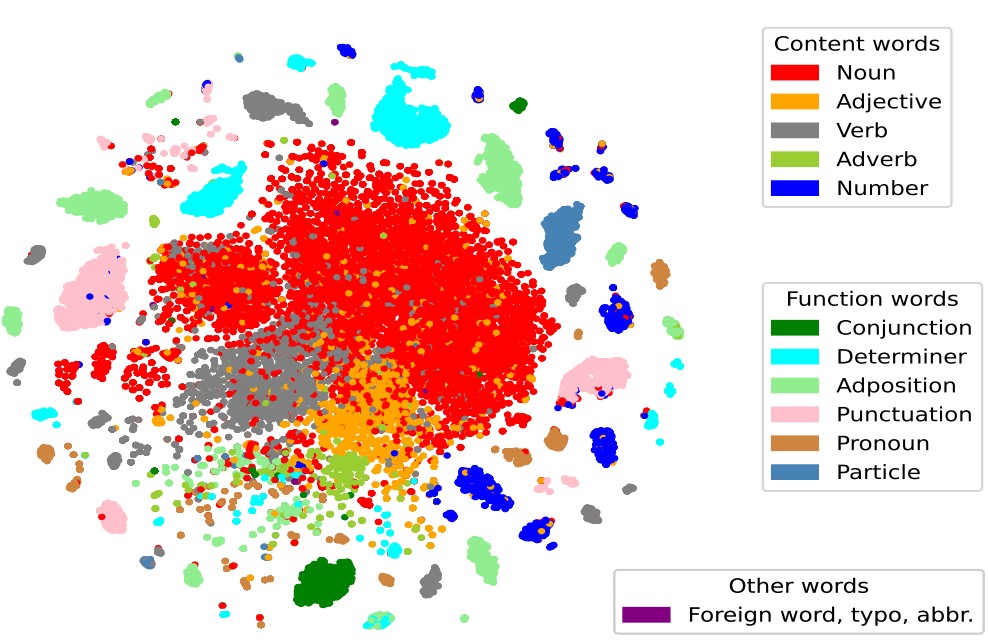

Figure 15: t-SNE of scores assigned by token embeddings in Qwen, colored by POS.

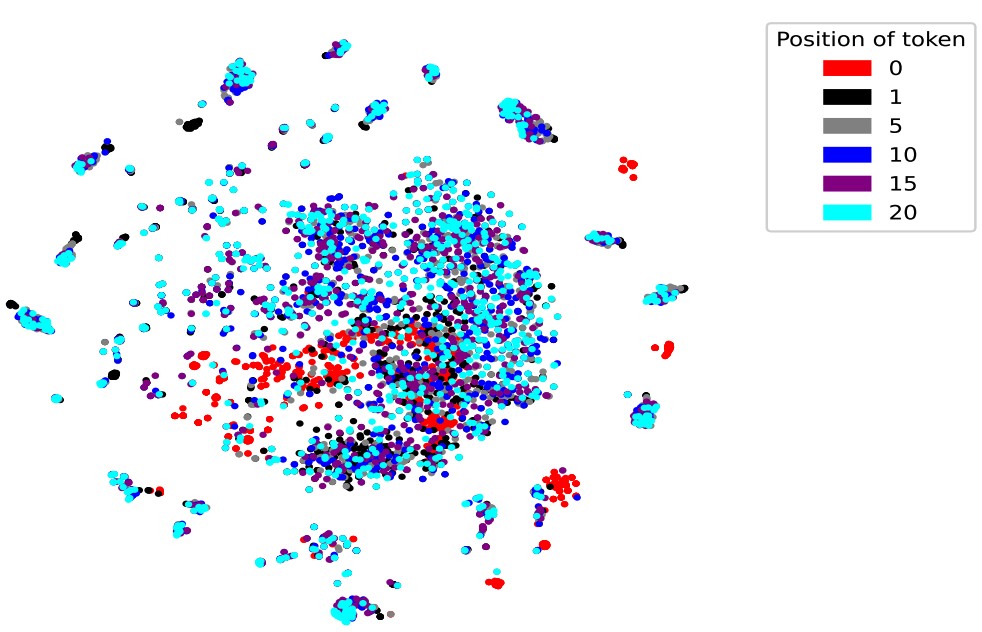

Figure 16: t-SNE of scores assigned by token embeddings in Qwen, colored by position.

# F  RESULTS ON IOI TASK

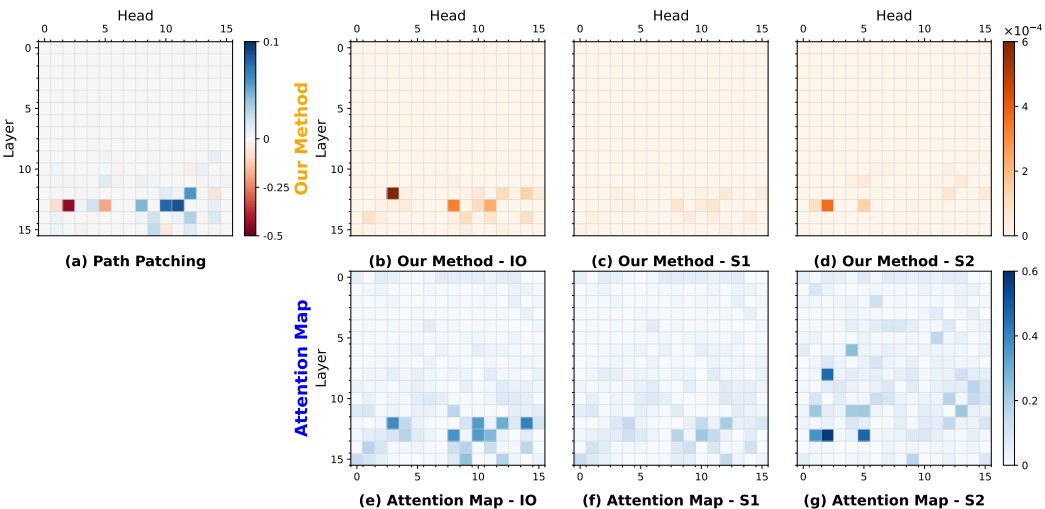

Figure 17: Path patching: (a) IO to logits (logit difference). Variance of scores assigned by: (b) (Head, Query=END, Key=S2), (c) (Head, END, S1), (d) (Head, END, IO) to the experts in the last block. Attention map: (e) (Query=END, Key=S2), (f) (END, S1), (g) (END, IO).

We follow Wang et al. (2023) to use "IO" to denote the indirect object, "S1" and "S2" denote the first and second occurrences of the subject, "END" denote the last token of the prompt. Additional details are provided in Appendix C.

We apply path patching for the pathway $h \to$ logits at END token for each head $h$ and compute the variation of logit difference $logit(IO) - logit(S)$, which is shown in Figure 17a. We observe that the Name Mover Heads (e.g., A12H12, A13H8, A13H10, A13H11) and Negative Name Mover Heads (A13H1, A13H2, A13H5) have a pronounced influence on the variation of logit difference, compared with other heads in the same layer. Since these function heads directly focus on the name tokens, we speculate that they have a relatively larger influence on the routing decisions through some specific terms, i.e., $\overline{\mathrm{LN}}_i^{\ell}(\boldsymbol{W}_O^{c,h}\boldsymbol{A}_{i,p}^{c,h}\boldsymbol{v}_p^{c,h})$ (Equation 9), where query $\boldsymbol{t}_i = $ END, key $\boldsymbol{t}_p \in \{IO, S1, S2\}$.

Given that the identified function heads have a high influence on the logits, we observe the variance of scores assigned to the experts in M15 (the last MoE layer) by tuples (head=AxHy, query=$\boldsymbol{t}_i$, key=$\boldsymbol{t}_p$), i.e., terms $\overline{\mathrm{LN}}_i^{15}(\boldsymbol{W}_O^{x,y}\boldsymbol{A}_{i,p}^{x,y}\boldsymbol{v}_p^{x,y})$ as an example. We find that the identified function heads tend to more influence the routing decisions, compared with the idle heads. The Name Mover Heads exhibit an apparent "activation" on [query=END, key=IO] and a very weak "activation" on [END, S1] (Figures 17b and c), indicating the scores assigned by tuples (A12H12, END, IO), (A13H8, END, IO), (A13H10, END, IO), and (A13H11, END, IO) have a comparatively high influence on the routing decisions in M15. We also identify a Name Mover Heads (A12H10) that have a weaker influence on both IOI circuits (Figure 17a) and routing decisions (Figure 17b), which could be a Backup Name Mover Head. In contrast, the Negative Name Mover Heads influence more in the variance score maps of [END, S2] (Figure 17d). A12H3 is noticeable for its influence on routing decisions (Figure 17b), which is likely to be a S-inhibition head as it assigns higher attention weight on IO instead of S1 and S2 (Figures 17e~g).

Finally, we compare the score variance maps (Figure 17b~d) with the attention maps (Figure 17e~g). We note that they resemble each other, particularly for the function heads. Although attention maps have some small non-zero activations in the early and intermediate layers, they usually do not have a strong contribution on the routing decisions in the last layers, which may due to the natural influence decay in deep layers, or the value vector cancels the effect. In conclusion, the function heads tend to have a comparatively noticeable influence the variance of assignment scores, which correlates with the corresponding attention map.

## G Results on scoring of attention heads in DeepSeek

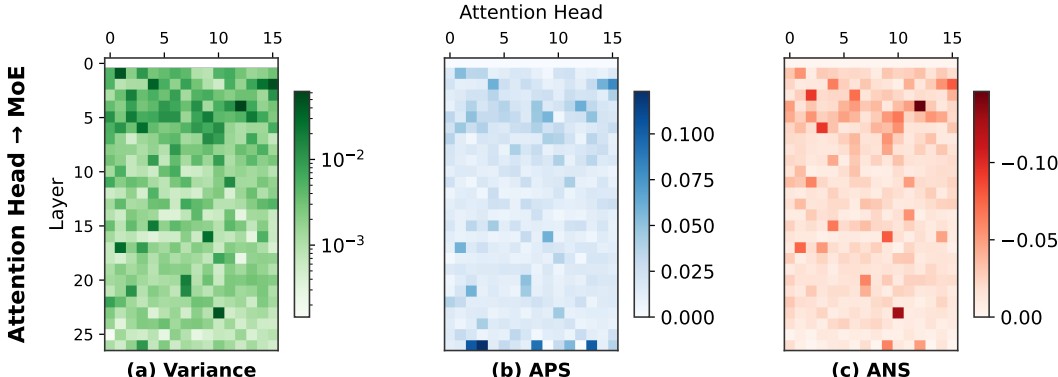

Figure 18: DeepSeek: (a) Variance of scores assigned by heads to experts in the same block. (b) Average positive scores (APS) of the heads. (c) Average negative scores (ANS) of the heads. Please note that Layer 0 of DeepSeek is an FFN layer, not an MoE layer.

## H Supplementary results on scoring of experts. results

Table 2: The rank of variance of scores assigned by the Super Experts to the experts in the next MoE layers.

| Model | Experts |
| --- | --- |
| DeepSeek | M2E54 (#55), M3H38 (#7) |
| OLMoE | Not Available |
| Qwen | M1H68 (#1), M2E92 (#3), M3E82 (#2) |
| Mixtral | M1E3 (#8) |

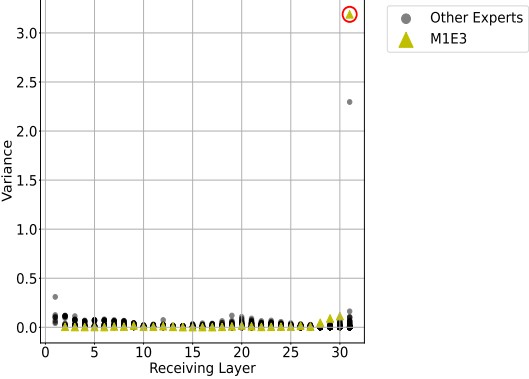

Figure 19: In Mixtral, M1E3 is found to be a "Super Expert", but it has a high variance at the top layers, especially the last layer (circled in red).

# I  ABLATION STUDY OF M1E9, M4E14, AND M1E18 IN OLMOE

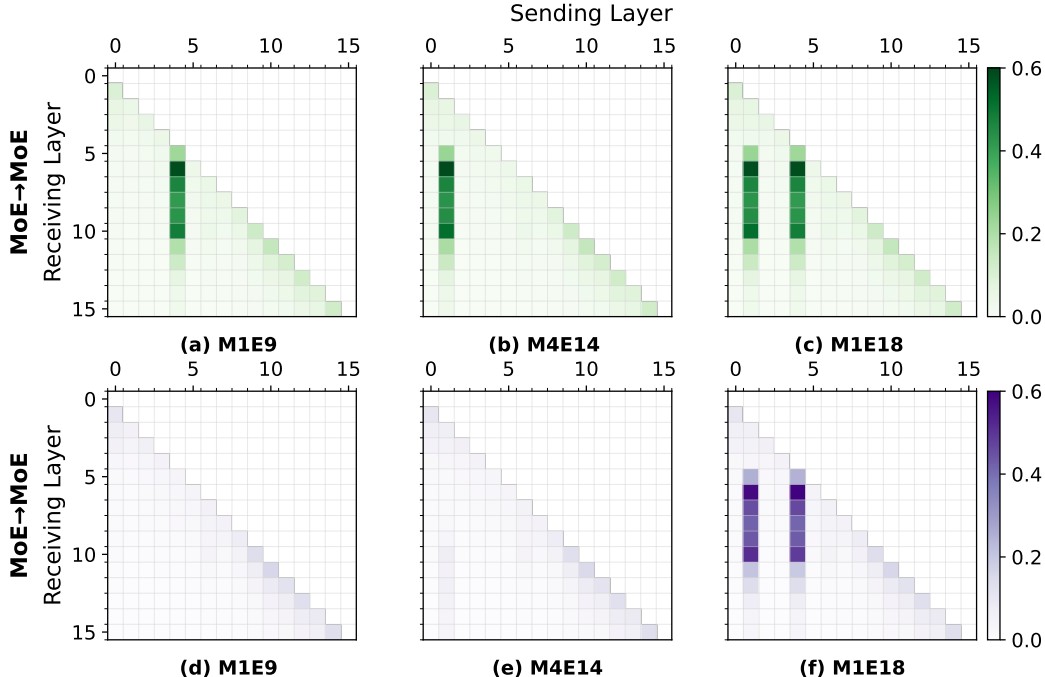

Figure 20: Variance of scores assigned by MoE Layer x to MoE Layer y. After setting the scores assigned by the specific expert to zero to perturb the routing decisions, we pad the output of MoE layers with original output: (a) M1E9, (b) M4E14, (c) M1E18. We do not perturb the routing decisions but pad the output of the specific expert as zero: (d) M1E9, (e) M4E14, (f) M1E18.

In this section, we conduct an zero-ablation experiment on Experts M1E9, M4E14, M1E18 in OLMoE as a case study to observe the influence on the routing decisions of M1E9 and M4E14, which are identified as large-variance experts in Figure 5a. M1E18 is the expert with a second large variance in the score assignment from M1 to M10, which is used for comparison.

We first set the scores assigned by the specific expert to zeros to observe how it perturbs the variance, and then pad the MoE layer output with the original output (Figure 20a∼c). In Figure 20a, we observe that the stripe at sending layer M1 vanishes after we set the scores assigned by M1E9 to zeros. Similarly, in Figure 20b, the stripe at sending layer M4 vanishes after we set the scores assigned by M4E14 as zeros. However, after we set the score to zeros assigned by M1E18, the two stripes persist (Figure 20c). The results indicate that both M1E9 and M4E14 have a significant influence on the contribution of M1 and M4 to the routing decisions in the layers in the stripe, respectively.

Now we do not perturb the routing decisions but just pad the output of the specific expert as zero vector (Figure 20d∼f). In Figure 20d, both stripes vanish after we apply the operation on M1E9, which indicate that the contribution of M4 is significantly influenced by M1E9, not just the contribution of M1. If we apply the operation on M4E14, we can observe that there is only a faint stripe corresponding to M1 (Figure 20e). We suggest that the output of M4E14 support the contribution of M1E9, which may also attribute to the attention layer 5 or the layer normalization operation after it. In Figure 20f, we again observe that the two stripes persist after applying the operation on M1E18, which indicates M1E18 does not have an apparent influence on the contribution of M1 and M4, compared with M1E9 and M4E14.

We summarize that: M1E9 first activates M4E14, and then they need to coexist to exert a significant influence on the following layers. This also explain why the stripes start from MoE Layer 5.

## J    RESULTS ON MATH REASONING TASK

In this section, we provide the experimental results on a subset of OpenR1-Math-220k (HuggingFace, 2025), a math reasoning task dataset, to investigate the generalizability of the scoring distribution pattern found in the main text. The results (Figures 21, 22, and 23) reveal that the scoring distribution pattern is consistent across different English corpora, as shown by comparing with the results in Figures 3, 4, and 5, with a minor change in the magnitude.

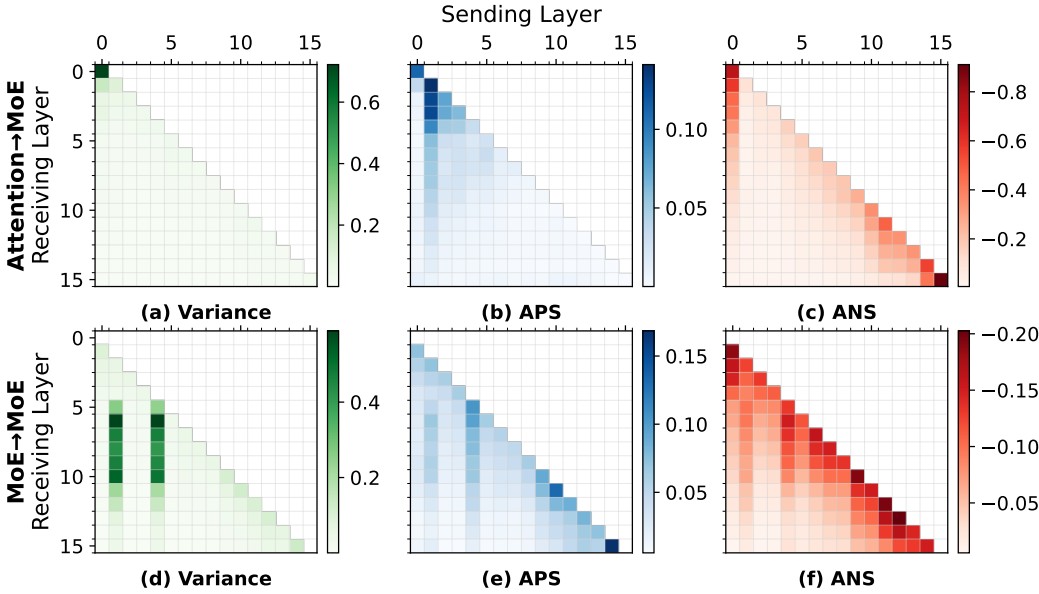

Figure 21: (Math reasoning task) Scores assigned by attention Layer x to MoE Layer y: (a) Variance, (b) Average positive scores (APS), (c) Average negative scores (ANS); Scores assigned by MoE Layer x to MoE Layer y: (d) Variance, (e) APS, (f) ANS.

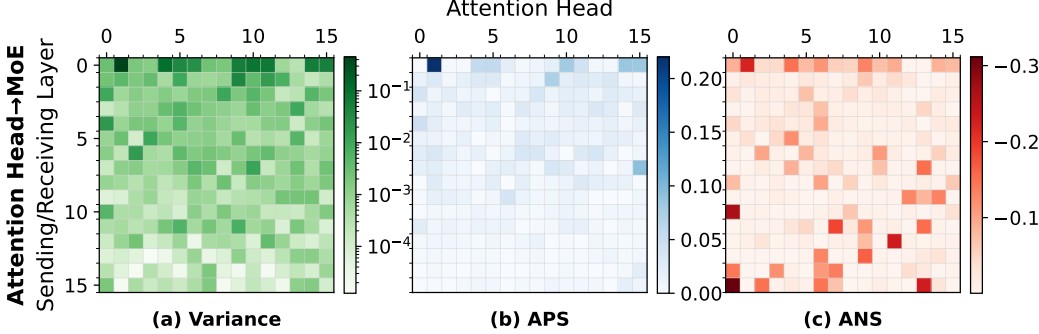

Figure 22: (Math reasoning task) Scores assigned by attention Layer x to MoE Layer y: (a) Variance of scores assigned by heads to experts in the same block. (b) Average positive scores (APS) of the heads. (c) Average negative scores (ANS) of the heads.

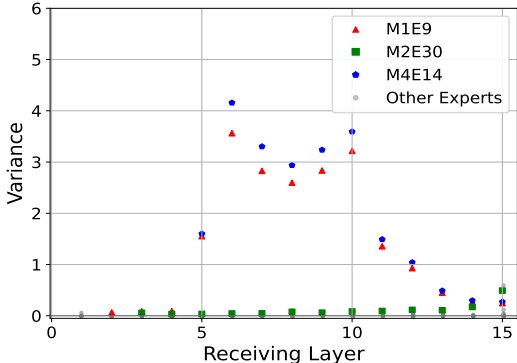

Figure 23: (Math reasoning task) The variance of scores assigned by experts to the following layers.

## K  L2-NORM HEATMAP

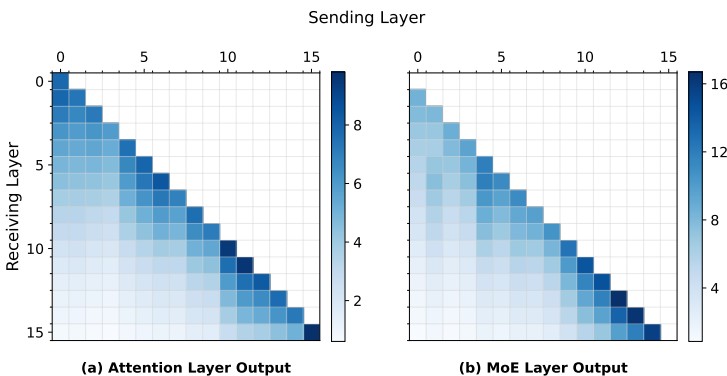

(a) Attention Layer Output          (b) MoE Layer Output

Figure 24: The L2-norm of projected sending layer output in different receiving layers. (a) attention layer output, (b) MoE layer output.

To study the difference of L2-norm of a component and the variance of scores assigned by the component, we illustrate the L2-norm of components in the general experiment on C4 dataset in Figure 24. In Figure 24a, the norm of projected sending attention layer output gradually decreases in the deeper receiving MoE layers. In Figure 24b, we again observe the existence of stripes at receiving layers M1 and M4. Both Figure 24a and b reveals that the large norm tend to occur in the deep sending layers. By Comparing Figure 24 and 3a and d, we conclude that a projected component with a large norm does not necessarily to have a high scoring variance.

## L  SENSITIVITY TO HYPERPARAMETER K (OF THE TOP-K)

In this section, we show that the cross-layer contribution is not sensitive to the hyperparemter K in OLMoE (original K=8, 64 experts in total) (Figure 25). We change K and rerun on C4 dataset with the same setting adopted in the main text. The pattern of score variance persists when K is changed, although the magnitude varies slightly.

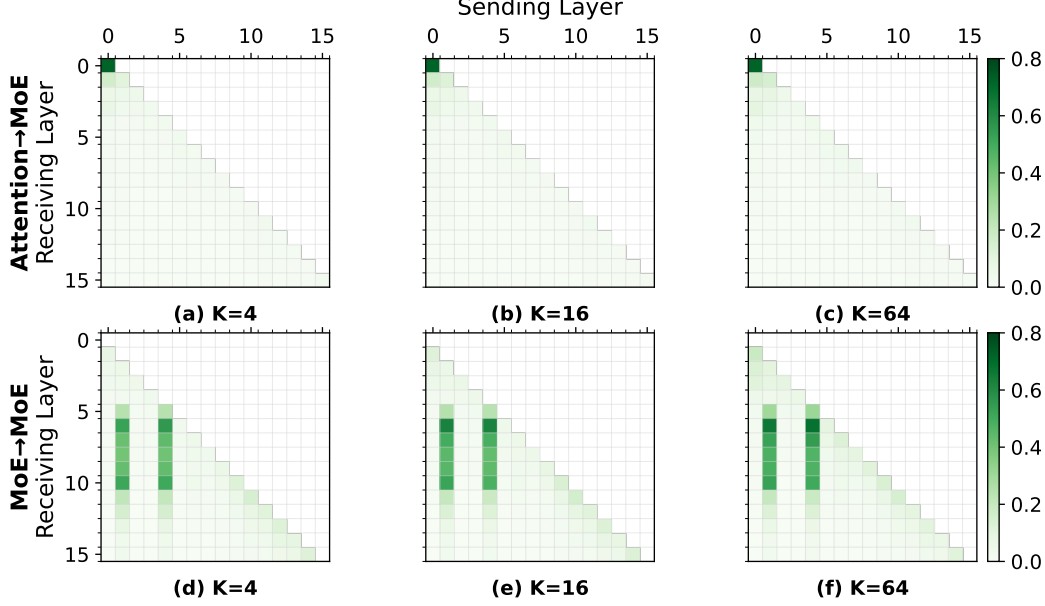

Figure 25: Variance of scores assigned by attention Layer x to MoE Layer y: (a) K=4, (b) K=16, (c) K=64; Variance of scores assigned by MoE Layer x to MoE Layer y: (d) K=4, (e) K=16, (f) K=64.

# M   GAP BETWEEN EXPERTS OF RANK K AND RANK (K+1)

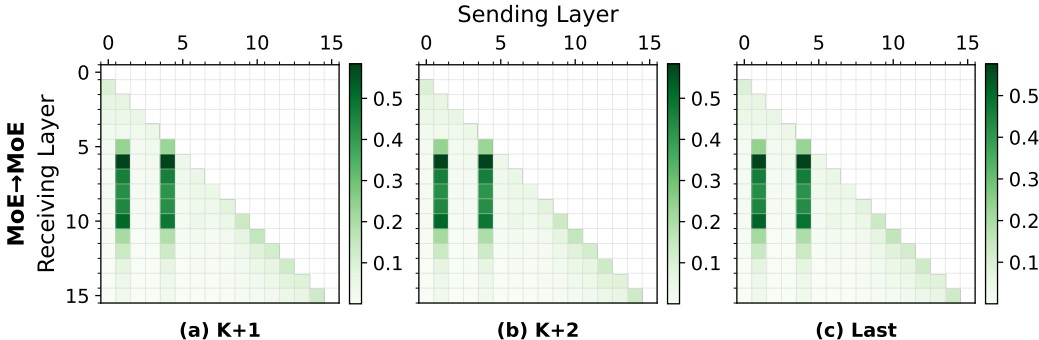

Figure 26: (Swapping rank K expert for an unselected expert) Variance of scores assigned by MoE layer x to MoE layer y: (a) K+1, (b) K+2, (c) Last ranked expert.

In this section, we show that the score gap between experts of rank K and K+1 is not evidently important for the routing decisions. We again conduct the experiment on C4 dataset but swap the rank the expert of rank K for the expert of rank K+1, K+2 or the last ranked expert (rank 64) during the routing decision (Figure 26) in OLMoE, and observe the impact on the variance. We can find that the gap between rank K and K+1 (or other unselected experts) has a fairly weak impact in the routing decisions in the following MoE layers. We suggest that the redundancy of selected experts lead to this phenomenon.

# N   RESULT OF SPEARMAN'S RANK CORRELATION

In this section, we give the Spearman coefficient of the cases we have studied in Section 7.2 (Figure 6). We measure the monotonic association between the variance of a component's score contributions and AARV of the receiving layer (defined in Section 7.2) using Spearman's rank correlation. We

observe Spearman coefficients of 0.595 (M1→M5), 0.708 (M1→M10), and 0.580 (M1→M15). These results indicate a moderate to strong monotonic correlation, with the correlation peaking at 0.708 for M1→M10).

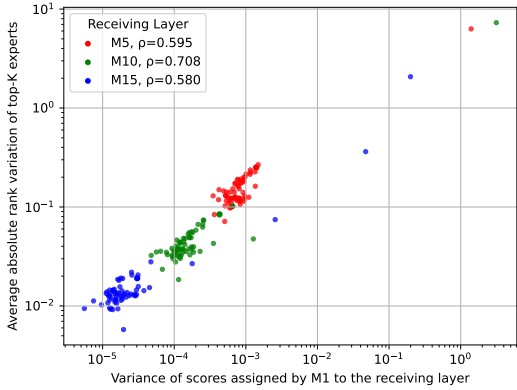

Figure 27: The variance of scores assigned by experts in M1 and the averange absolute rank variation of top-K experts. The Spearsman coefficient $\rho$ is shown in the legend, all the p-values $< 0.001$.

## O  MODEL DESIGN IMPLICATIONS

a) **Non-uniform MoE placement**. Our variance maps show that a few MoE layers consistently dominate the routing landscape, while others have comparatively modest contribution. This suggests exploring heterogeneous depthwise allocation, where MoE capacity (number of layers, width, or top-K) is concentrated in high-influence regions, and reduced or even removed in consistently low-influence regions.

b) **Tiered expert capacity**. At the expert level, we observe that only a small subset of experts maintains strong contributions across many layers. This motivates architectures with tiered experts—for example, a mix of "high-capacity" experts in high-influence layers and cheaper experts elsewhere, rather than a uniform design.

c) **Cross-layer–aware routing heuristics**. Since routing decisions are not purely local, one could use our cross-layer contribution maps as a prior for inference-time scheduling: e.g., prefetching experts that are systematically promoted by influential upstream layers, or biasing training-time regularizers to stabilize or diversify those long-range pathways.

## P  A BRIEF MECHANISTIC EXPLANATION OF THE OBSERVED PHENOMENA

a) Why MoE outputs persist more than attention outputs? MoE FFN outputs have larger post-normalization magnitudes and expert-specific projections. Since router scores are dot products with normalized inputs, this gives MoE components a much larger capacity to introduce variance across experts. Attention outputs are redistributive and tend to decay across depth, leading to more local effects.

b) Why "entanglement stripes" appear? Once a high-impact expert fires in layer $\ell$, its output is repeatedly re-projected through subsequent blocks. When this direction aligns with router weight vectors in deeper layers, the influence accumulates rather than dissipates. In addition, the results indicate that router weights across layers are not random but partially aligned, creating shared routing subspaces and the observed stripes.

c) Why cross-layer coupling is strong in MoEs? Top-K sparsity stabilizes early expert selections, and expert-specific nonlinear transformations generate distinctive activation signatures that persist across depth. These signatures repeatedly bias downstream routing. In contrast, attention layers mainly remix existing signals and do not introduce expert-specific basis directions, which explains their weaker long-range persistence.

