# OpenReview forum: "Understanding Cross-layer Contributions to Mixture-of-Experts Routing in LLMs"
_ICLR.cc/2026/Conference — ICLR 2026 Poster_

### Official Review · Reviewer_NVZU · 2025-10-28

**Soundness:** 3
**Presentation:** 2
**Contribution:** 2
**Rating:** 4
**Confidence:** 4

**Summary:**

This paper investigates the routing mechanisms in Mixture-of-Experts (MoE) LLMs from a cross-layer perspective. The core idea is to recursively decompose the expert assignment score, i.e., the result of a dot product between an expert's routing weight vector and the MoE layer's input, into a linear sum of contributions from all preceding model components. These components include the initial token embedding, the outputs of all previous attention layers, and the outputs of all previous MoE layers.

The authors define a metric for "influence" based on the variance of the scores a single component contributes to all experts in a receiving MoE layer; high variance implies a strong influence on the routing decision. Using this framework, the paper analyzes OLMOE, DeepSeek-V2-Lite, Qwen3-30B-A3B, and Mixtral-8x7B.  Here are some of their key findings:

The variance of the initial token embedding scores decays rapidly with layer depth.

The authors identify a "cross-layer entanglement" phenomenon, where specific MoE layers exhibit a strong, persistent, long-range influence on routing decisions in many subsequent layers.

The methodology could be extended to finer granularities, showing that specific attention heads or a small subset of experts are disproportionately influential on routing.

**Strengths:**

- Intuitive and straightforward method: The paper presents a simple and computationally light method for MoE interpretability. Shifting the focus from expert specialization (what experts do) to the composition of the router's input (why a router chooses) is a good contribution to the field.

- Interesting Findings: The findings of this paper clearly demonstrate that MoE routing is not just shaped by the precedent attention layer but also previous transformer blocks.

**Weaknesses:**

1. **My greatest concern** for this paper is whether variance is a good metric for an expert’s contribution. The paper's metric (variance) measures "flatness" of gating scores, but this is an imperfect proxy for influence on the Top-K routing decision (expert selection and weight). For example, a component's contribution that changes scores (assuming selecting 2 experts out of 4 experts) from [10, 9, 1, 0] to [12, 7, 2, -1] with contribution [2, -2, 1, -1] (Top-K={E1, E2}) would have a high variance. A different component that changes scores from [5, 4.9, 4.8, 4.7] to [6, 3.9, 4.8, 4.7] (Top-K={E1, E3, a changed routing decision) with contribution[1, -1, 0, 0]. The latter alter the routing results but have lower variance.  The metric is not sensitive to the "gaps" between the K-th and (K+1)-th expert, which is what actually determines the decision. Mean while, the variance also does not reflect the final contribution/weight of each expert. For example, a component contribute [100, 100, 100, 100] to the gating score and another component contribute [1, 1, 0, 0]. The first component’s contribution is zero accroding to variance. but it actually make the final selective expert’s contribution more flat, which is quite important and cannot be ignored when understanding routing mechanism in MoE.

2. **Does this paper re-find the massive activation phenomenon?** The observations discovered by authors closely relates to massive activation and attention sink. I hope authors could discuss the relation with these two phenomenons.  Furthermore, the author should to provide the most direct piece of evidence for illustrating the contribution of each components: a heatmap of component L2-norms, directly comparable to Figure 3. The findings in Figure 3 (e.g., high influence from M1, M4) could very well be a simple re-discovery of massive activation of that layer. Without this comparison, the paper's core claim of novelty is undefended.

3. **Fragmented Narrative**: I feel that the paper's presentation is highly fragmented. This is a major weakness because it buries the paper's core discovery. Making readers easy to conclude that the findings are just a restatement of known concepts (i.e., massive activation). By the way, the title in the PDF is different from the submission.

4. **Speculative Practical Applications**: The abstract and conclusion claim the findings "highlight the opportunities... for effective model design and model serving." However, these claims are speculative and undeveloped.

**Questions:**

1. Can the authors provide a layer-level heatmap (identical in structure to Figure 3) but plotting the average L2-norm of the component outputs instead of the score variance? If this norm-heatmap is nearly identical to the variance-heatmap, the paper's claim to novelty is significantly weakened.

2. Could the authors comment on the limitations of variance as a metric, specifically its insensitivity to the score "gaps" between the Top-K and non-Top-K experts and selected expert’s contribution? Do we need additional metrics for this?

3. Assuming the variance metric is not just a proxy for norm (as per Question 1), what is the authors' primary hypothesis for the observed divergence between high-norm and high-influence components?

4. Have the authors investigated whether these cross-layer influence patterns remain consistent during autoregressive generation on specific tasks (e.g., question-answering or MATH reasoning)? Is it possible that the observed routing patterns are an artifact of the C4 dataset and the forward-pass-only analysis?

---

> ### Author Response · Authors · 2025-11-24
> **Official Comment by Authors (Part 1)**
>
> We thank you for your time and constructive feedback. We address the raised weaknesses and questions below.
>
> ## Response to the weaknesses:
>
> ### W1. Is variance a good metric for contribution?
>
> **(Copied from General Response 2)** A routing decision is determined by the scores of all experts. Hence, we apply the variance of scores assigned by a component to all the experts in an MoE layer to measure how the component influences the routing decisions. The intuition is that a higher variance leads to a more uneven distribution of assignment scores, hence the corresponding component has a higher influence on the routing decisions. We have now added an experiment to demonstrate that variance of the assignment score of a component can reflect its impact on the ranking of top-K experts in a certain degree (Section 7.2).
>
> **(Additional Response)** We acknowledge the reviewer's concerns regarding whether variance perfectly reflects the routing impact. Variance is not intended to capture every possible notion of “importance”; it captures one specific aspect that directly follows from the router’s scoring operation. We intend to use variance to measure the influence of a component on the routing decision of all the experts in an MoE layer, not just selected experts. From Proposition 1 and the form of the dot product, any component with zero variance is guaranteed to have zero impact on ranking changes. In this sense, variance provides a necessary condition for influence.
>
> We now discuss the limitations: variance does not account for score gaps near the Top-K boundary; variance does not indicate whether the routing actually changes; variance does not represent final expert weights after softmax. Despite these limitations, we chose variance because it is model-agnostic, causally monotone with respect to scaling a component, and works uniformly across all granularities (token, head, expert, layer). We deeply appreciate the reviewer’s suggestion of exploring more complex Top-k-aware metrics; we will include this as future work.
>
> ### W2: Does this paper re-find the massive activation phenomenon?
>
> **(Copied from General Response 4)** Super experts (Su et al.) are characterized by having unusually large output magnitude, which implies they can have a strong impact on the model’s final output. This is an observation about representational strength at the level of model outputs.
> In contrast, our metric is designed to characterize cross-layer (both short- and long-range) correlations by quantifying how each component contributes to the expert score landscape. Rather than focusing on output dominance, our method serves a different purpose: it enables fine-grained mechanistic analysis of how routing signals are shaped and propagated across layers.
> In relation to the above, we also would like to emphasize that the result in Figure 5 (and line 472) in the paper demonstrate that not all the Super Experts have typically large scoring variance, i.e. are not necessarily contributing highly to all following layers.This indicates that the set of Super Experts and the large-variance experts found by our method are different although they may have an overlap.
>
> **(Additional Response)** High-influence layers often also have high norms, but the match is imperfect. In this work, we show that the stripes are triggered by several experts (e.g., M1E9, M4E14 in OLMoE). Furthermore, we observe that the norm of these experts' output is the largest among the experts in their layers. However, we suggest that a large norm is not a sufficient condition for experts with such unusual behavior. We show the average L2-norm of some experts' output in OLMoE below (this is not included in the paper):
>
> | Expert | M1E9 | M1E18 | M2E30 | M3E39 | M4E14 |
> | - | - | - | - | - | - |
> |L2-norm | 0.78 | 0.72 | 12.89 | 1.61 | 0.87 |
>
> We notice that although M2E30 has a significant norm, it does not have a proportionately strong influence on the routing decisions in most following layers. M1E9 and M1E18 are two outliers in MoE Layer 1, however, only M1E9 has an apparent influence. And we also observe that such experts occur in the intermediate layers in Qwen. These experts are inconsistent with the "Massive activation" and "Super Experts" which indicate that the activations/experts they found are in the early layers. Hence, we consider that there is a difference between the existing findings on "Massive activation" and our findings.
>
> ### W3: Fragmented Narratives
>
> Thank you for the reminder. We will improve the writing. We apologize for the inconsistency between the title on the webpage and the PDF file, but the former cannot be changed at this stage. Please refer to the title in the PDF file.
>
> ### W4: Speculative Practical Applications
>
> We will adjust the abstract and conclusion to avoid sounding speculative. We now frame the applications more modestly, emphasizing that our influence maps enable rather than provide design strategies.

---

> ### Author Response · Authors · 2025-11-24
> **Official Comment by Authors (Part 2)**
>
> ## Response to the questions:
>
> ### Q1, Q3: Average L2-norm heatmap
>
> We have now added a figure to show the L2-norm of layer outputs in Appendix K (Figure 24). We prove that the L2-norm of a component influences the *upper bound* of the variance of scores assigned by it mathematically in Appendix A.
> The result and proof demonstrate that the L2-norm alone is not capable of giving the actual contribution, as the variance metric does; the L2-norm is only capable of giving the upper bound on the contribution.
>
> ### Q2. Variance as a metric for contribution
>
> **Regarding the gaps between experts:** In this work, we mainly focus on how a component influences the routing decisions separately. However, the gaps between experts are determined by all the previous components. Since these gaps are contributed by many components, they can be so fuzzy that one may even select Expert (K+1) instead of Expert K without apparently impairing the performance, especially when the number of experts is large. We suggest that some gaps are obvious and important, such as the gap between rank 1 and rank 2, as they may get a high rank due to the high scores assigned by several components; while some gaps are not, and these gaps are usually not clear enough to understand (i.e., attributed to several components clearly). \ccc{We have now added an experiment to show that even when the rank K expert is swapped with rank K+1, it does not significantly change the pattern of variance of scoring (Appendix M, Figure 26).}
>
> **Regarding the contribution to the selected expert:** We recognize that the metric adopted in this work does not reflect all types of "contribution", such as "flattening" the weight of selected experts, as mentioned by the reviewer. We suggest that this type of "contribution" can be measured using a metric that considers both magnitude and the flatness of assigned scores. A possible metric can be $\frac{\mu}{Var}$, where $\mu$ is the average assigned score, $Var$ is the variance.
>
> Furthermore, we also provide a new metric in Section 7.2, i.e., the average absolute rank variation of top-K experts, to measure how the top-K experts in an MoE layer would change if the scores assigned by a previous component (e.g., an expert) to the experts in that layers are set to zero.
>
> ### Q4. Are the observed phenomena consistent across different tasks?
>
> **(Copied from General Response 5)** We have now added experimental results on a math reasoning task dataset (Appendix J), which reveals a similar pattern of scoring (e.g., stripes and high-influence experts) that occurs in the C4 dataset.
>
> **(Additional response)** While we acknowledge that these phenomena may vary on tasks not tested, we suggest these patterns are general features of English text processing, given the breadth of the C4 dataset and the persistence of the patterns in the math reasoning task dataset.

---

> ### Comment · Reviewer_NVZU · 2025-11-27
>
> Thanks for the author's effort to address my concern. I hope authors could keep improving the writing for a more systematic presentation and to highlight the paper's main contribution. I've raised my score accordingly.

---

> ### Author Response · Authors · 2025-11-27
>
> Thank you!
>
> We would be glad to answer any more questions about the paper.

---

### Official Review · Reviewer_T74p · 2025-10-29

**Soundness:** 2
**Presentation:** 2
**Contribution:** 2
**Rating:** 4
**Confidence:** 4

**Summary:**

This paper investigates the routing mechanism in MoE-LLMs from a cross-layer mechanistic interpretability perspective. The authors propose a lightweight, recursive decomposition methodology to attribute the routing score assigned to each expert back to the contributions of preceding model components (token embeddings, attention layers/heads, and MoE layers/experts). Applying this methodology to four different large-scale MoE models (OLMOE, DeepSeek-V2-Lite, Qwen3, Mixtral), the study identifies several common patterns: (1) MoE layer outputs generally contribute more significantly and persistently to downstream routing decisions compared to attention layer outputs; (2) Evidence of "MoE entanglement," where the activation of experts in earlier layers correlates with the activation of experts in later layers; (3) Certain components exhibit long-range influence, affecting routing decisions many layers downstream; (4) Models exhibit distinct patterns regarding short-range versus long-range promoting or inhibiting effects from components on subsequent routing.

**Strengths:**

1. The paper addresses a timely and important gap in understanding MoE models by focusing on the cross-layer dynamics influencing routing decisions.
2. The empirical results reveal non-trivial cross-layer interactions, such as the stronger, persistent influence of MoE outputs compared to attention outputs, the "entanglement" effect, and long-range influences. These findings challenge simplistic, localized views of routing and provide valuable insights into the complex dynamics within these models.

**Weaknesses:**

1. Equation 8 uses an approximation ($\overline{LN}_{i}^{l}(z)$) where the RMS term is calculated over the *total* layer input, but the contribution of a *single* component $z$ is scaled. Given the non-linear nature of RMSNorm (depending on the norm of the entire input vector), the accuracy and potential biases introduced by this approximation are unclear and not quantified. This could affect the relative attribution scores between components.

2. While the paper successfully identifies several interesting cross-layer patterns (e.g., MoE persistence, entanglement stripes), it provides limited mechanistic explanations for *why* these patterns emerge. For instance, what structural or functional properties of MoE layers enable their outputs to have a more persistent influence compared to attention layers? What specific information pathways lead to the observed "entanglement" between expert activations across layers?

3. The paper could be clearer about how metrics like variance, APS, and ANS are aggregated (e.g., averaged across which tokens, which samples?) when presenting layer-level heatmaps (e.g., Figure 3).

**Questions:**

1. Could you elaborate on the validity of the $\overline{LN}_{i}^{l}(z)$ approximation used in Equation 8, particularly for RMSNorm?

2. Regarding Proposition 1, how strongly does the variance of a component's score contributions correlate with its actual impact on changing the set of top-k selected experts?

3. You suggest using cross-layer contribution insights for expert prefetching. Could you sketch a potential algorithm or strategy for how the variance or entanglement patterns identified could be used to predict which experts are likely to be needed in future layers, thereby informing a prefetching policy?

**Details Of Ethics Concerns:**

I did not find Section "THE USE OF LARGE LANGUAGE MODELS" in this paper. This would violate ICLR submission rules and should be desk-rejected.

---

> ### Author Response · Authors · 2025-11-24
> **Official Comment by Authors (Part 1)**
>
> We apologize for the mistake of omitting the section "The Use of Large Language Models". We have strictly adhered to ICLR guidelines and have added this required section in Section 9. We thank you for your time and constructive feedback. We address the raised weakness and questions below.
>
> ## Response to the weaknesses:
>
> ### W1: Regarding the approximation
>
> **(Copied from General Response 1)** Our decomposition uses $\mathrm{\overline{LN}}(c)= \frac{\gamma \odot c}{\mathrm{RMS}(c_{total})}$, where $c$ is a component vector and the denominator depends on the full input vector $c_{total}$. Therefore, the decomposition distributes the \textbf{same normalization factor} across all components. This preserves the direction of each component's contribution and keeps their relative magnitudes faithful to what the router actually receives. The decomposition should be viewed as **"what each component looks like after the shared normalization factor has been applied,"** rather than **"the output of LayerNorm applied to that component in isolation."**
>
> Furthermore, we have now added a case study of direct causal analysis in Section 7.2 to validate the effectiveness of the proposed decomposition method.
>
> **(Additional Response)** This operation $\mathrm{\overline{LN}}(c)$ is \textit{not} assuming linearity of RMSNorm; it simply isolates the linear term that the router weights interact with during the dot product. We do not intend to measure the whole non-linear influence of a component $c$ (e.g., $\mathrm{RMSNorm}(c_{total}) - \mathrm{RMSNorm}(c_{total} - c)$, where $c_{total}$ is the original input vector of RMSNorm) as there is no absolute "ground truth" for it.
>
> ### W2: Lack of a mechanistic explanation of the observed phenomena
>
> We agree this is important and have added a short discussion in Appendix P. In brief, our analysis suggests three structural factors:
>
> a) Why MoE outputs persist more than attention outputs? MoE FFN outputs have larger post-normalization magnitudes and expert-specific projections. Since router scores are dot products with normalized inputs, this gives MoE components a much larger capacity to introduce variance across experts. Attention outputs are redistributive and tend to decay across depth, leading to more local effects.
>
> b) Why "entanglement stripes" appear?
> Once a high-impact expert fires in layer $\ell$, its output is repeatedly re-projected through subsequent blocks. When this direction aligns with router weight vectors in deeper layers, the influence accumulates rather than dissipates. In addition, the results indicate that router weights across layers are not random but partially aligned, creating shared routing subspaces and the observed stripes.
>
> c) Why cross-layer coupling is strong in MoEs? Top-K sparsity stabilizes early expert selections, and expert-specific nonlinear transformations generate distinctive activation signatures that persist across depth. These signatures repeatedly bias downstream routing.
> In contrast, attention layers mainly remix existing signals and do not introduce expert-specific basis directions, which explains their weaker long-range persistence.
>
> Additionally, we have now added a zero ablation study on experts M1E9 and M4E14 in Appendix I which reveals that M1E9 first activates M4E14 and the two experts coexist to exert an apparent influence on the routing decisions in the following layers. While a full mechanistic theory is outside the scope of this work, we believe these structural factors offer a concrete interpretation of the observed patterns.
>
> ## W3: Clarification on the aggregation of the used metrics
>
> We have now clarified the way how these metrics are aggregated in the paper (Page 5, Line 258). Variance, APS, ANS are computed per token position, then averaged over non-lead tokens, and then averaged over samples. Thank you for noticing that the text was not as explicit as it should have been.
>
> ## Response to the questions:
>
> ### Q1: Validity of the approximation
>
> Please refer to the response in W1. We do not intend to approximate a specific metric regarding "non-linear influence" as we consider that there is no such ideal "ground-truth".

---

> ### Author Response · Authors · 2025-11-24
> **Official Comment by Authors (Part 2)**
>
> **Nov 24: We edited the incorrect reported magnitude of Spearman coefficients. We apologize for the inconvenience.**
>
> ### Q2: Variance and the Top-K selected experts
> To quantify this relationship, we conducted an additional experiment (see Figure 27 in Appendix N). We measure the monotonic association between the variance of a component’s score contributions and its actual impact on altering the top-K expert set using Spearman’s rank correlation. We observe Spearman coefficients of 0.595 (M1→M5), 0.708 (M1→M10), and 0.580 (M1→M15). These results indicate a moderate to strong monotonic correlation, which becomes stronger as the model depth increases (peaking at 0.708 for M1→M10), suggesting that components with higher score variance tend to have a correspondingly larger influence on expert selection dynamics. This supports our claim that variance is a meaningful signal for understanding and predicting expert churn, particularly in deeper regimes where routing behavior stabilizes.
>
> ### Q3. Regarding the expert prefetching
>
> While we do not implement an expert prefetching system in the current paper, our analysis naturally suggests a concrete strategy for building one on top of the variance and entanglement structures we uncover.
>
> **Phase 1: Offline construction of a routing influence graph.**
>
> Using our decomposition framework, one can estimate cross-layer influence patterns between experts. For each layer pair $(\\ell, \\ell')$ and experts $e \\in MoE_\\ell$, $e' \\in MoE_{\\ell'}$, we define an influence measure such as the empirical conditional probability that expert $e'$ is selected in layer $\\ell'$ given that expert $e$ is highly contributing in layer $\\ell$.
>
> This yields a sparse, directed *routing influence graph* over experts. For scalability, this graph can be built at different granularities (e.g., expert clusters or layer-level aggregation) rather than at the level of individual experts.
>
> **Phase 2: Online expert prefetching policy.**
>
> At inference time, for each token (or micro-batch) and each MoE layer $\\ell$:
> - Let $E_{\\ell}^{active}$ denote the set of experts selected at layer $\\ell$.
> - Using the precomputed influence graph, estimate a predictive distribution over experts in a future layer $\\ell' > \ell$: $\\hat{P}(e' \\in Top-k_{\ell'}) \\approx f ( \\{ I_{\\ell \\rightarrow \\ell'}(e \\rightarrow e') |  e \\in E^{active}_\\ell \\} ),$
>
>     where $f$ can be a simple weighted aggregation followed by normalization.
> - The system then prefetches the top-$M$ experts with the highest predicted probabilities into fast memory or closer devices (e.g., GPU HBM or local cache) before layer $\\ell'$ executes its routing step.
>
> To ensure practicality, the policy can be conservative: prefetching only when predicted probabilities exceed a threshold, or only for those MoE layers that exhibit strong long-range entanglement in our analysis (e.g., the early influential layers observed in Fig. 3d).
>
> Conceptually, this approach treats our cross-layer variance and entanglement maps as a learned prior over future routing behavior, enabling more informed expert scheduling during inference. A full systems implementation is beyond the scope of this paper, but we will add a concise discussion of this potential application in the revised manuscript.

---

> ### Author Response · Authors · 2025-11-28
>
> Gentle reminder: we would be more than happy to answer any question, or clarify any point.

---

### Official Review · Reviewer_tjsZ · 2025-10-30

**Soundness:** 3
**Presentation:** 3
**Contribution:** 3
**Rating:** 6
**Confidence:** 3

**Summary:**

This paper proposes a lightweight recursive decomposition method for quantifying the cross-layer contributions of different model components to MoE routing decisions. Through experiments on four production-level MoE models, the authors find:

  -Dominant Role of MoE Layer Outputs: The outputs of MoE layers exert the strongest and most persistent influence on the routing decisions of subsequent layers.

  -Cross-Layer Entanglement: Routing decisions are influenced by long-range cross-layer interactions, challenging the assumption that routing is a local process.

  -Component Effects: Attention layer outputs exhibit local facilitative effects in the bottom and top layers, while MoE layer outputs demonstrate global suppression in the middle layers.

  -Impact of Key Experts and Attention Heads: A small number of super experts have a sustained impact on routing.
These findings highlight the importance of cross-layer interpretability and provide new opportunities for the efficient design of MoEs.

**Strengths:**

-Lightweight Design: The recursive decomposition method requires no architectural modifications or retraining, achieving decomposition solely through input manipulation with low computational overhead. This offers a new paradigm for MoE interpretability research.

  -Model Diversity: Covers four mainstream MoE architectures, validating the universality of the conclusions. Supplementary evidence in the appendix, including t-SNE clustering and heatmaps, provides intuitive visualization of the spatial and hierarchical distribution of component contributions.

  -Theoretical Rigor: Propositions 1 and 2 prove the rationality of using variance as an influence metric, establishing a mathematical foundation for the analysis.

  -Discovery of Super Experts and Long-Range Suppression: Identifies super experts and long-range suppression effects, offering new ideas for optimizing MoE systems.

**Weaknesses:**

-Limited Model Scope: Experiments are confined to open-source models, excluding systems like GPT-4 MoE or other closed-source implementations. The generalizability of conclusions to larger-scale or heterogeneous architectures remains questionable.

  -Task Simplicity: Only the C4 and IOI tasks are used, lacking validation on complex downstream tasks such as reasoning or multimodal tasks. This makes it difficult to prove the task invariance of routing patterns.

  -Missing Attribution Method Comparison: No comparison is made with gradient-based attribution methods or causal intervention methods. The advantages of the variance metric are not fully substantiated.

  -Sensitivity to Hyperparameters: The analysis does not examine the impact of routing hyperparameters like the Top-k value or number of layers on the conclusions. For instance, Mixtral's Top-k value of 2 might amplify locality effects.

**Questions:**

Same as weakness.

---

> ### Author Response · Authors · 2025-11-24
> **Official Comment by Authors**
>
> We thank you for your time and helpful feedback. We respond to the weaknesses pointed out below.
>
> ### W1: Limited Model Scope
>
> We have applied our method to four open-source models of different scales. We acknowledge that whether the generalization of the results in the paper is still true in larger models or heterogeneous architectures remains to be unclear. Like all work on mechanistic analysis, and not just this paper, it is not possible to conduct experiments on closed-source models since we do not have access to their weights. We hope that the method in this paper, once in the open space, could be used by developers of closed models.
>
> ### W2: Task Simplicity
>
> **(Copied from general response 5)** We have now added experimental results on a math reasoning task dataset (Appendix J), which reveals a similar pattern of scoring (e.g., stripes and high-influence experts) that occurs in the C4 dataset.
>
> **(Additional response)** While we acknowledge that these phenomena may vary on tasks not tested, we suggest these patterns are general features of English text processing, given the breadth of the C4 dataset and the persistence of the patterns in the math reasoning task dataset.
>
> ### W3: Missing Attribution Method Comparison
>
> **(Copied from General Response 3)** We have now added a case study of causal analysis in Section 7.2 and Appendix I. We show in a physical perspective that the expert M1E9 with the largest variance also has the highest influence on the top-K expert selections in following layers, compared with other experts in M1. Furthermore, we identify that M1E9 controls the activation of M4E14 and they need to coexist to exert a prominent influence on the routing decisions in MoE Layer 5 and the following layers. The results from the causal study confirm that these experts assigning high-variance scores tend to have a strong influence on the routing decisions and validate the effectiveness of our methods.
>
> ### W4: Sensitivity to Hyperparameters
>
> We have now added an experiment on the sensitivity to hyperparameters (Appendix L). We observe that the patterns of cross-layer effects (e.g., "stripes" and "high-variance experts") are not apparently sensitive to the hyperparameter K in OLMoE.

---

> ### Author Response · Authors · 2025-11-28
>
> Gentle reminder: we would be more than happy to answer any question, or clarify any point.

---

### Official Review · Reviewer_Dj7p · 2025-11-01

**Soundness:** 3
**Presentation:** 2
**Contribution:** 3
**Rating:** 6
**Confidence:** 4

**Summary:**

The paper analyzes the routing mechanism in Mixture-of-Experts (MoE) language models from a cross-layer mechanistic interpretability perspective.

The authors propose to understand routing by recursively decomposing the expert assignment score. The score for a given expert is a dot product of its weight vector and the MoE layer's input vector (Eq. 6). The paper's key insight is to decompose this input vector into a sum of its constituent parts: the original token embedding plus the outputs from all preceding attention and MoE layers (Eq. 8).

some of the key findings include:

- MoE layer outputs generally have a stronger and more persistent influence on downstream routing decisions than attention layer outputs.
- The paper finds that routing is not a purely local decision. Certain MoE layers (and specific experts within them, e.g., M1E9 and M4E14 in OLMOE) exhibit high influence on routing decisions many layers deeper, creating "stripes" of influence (visible in Fig 3d).

**Strengths:**

- The paper's primary strength is its shift in perspective. Moving from "what do experts specialize in?" to "what components influence the choice of expert?" is a significant contribution. The method of decomposing the router's input vector is elegant and interpretable.

- Clear and Justified Metrics: The use of variance as a proxy for "influence" (Proposition 1) and APS/ANS for "tendency" (Proposition 2) is well-defined and effective. This provides a clear, quantitative framework to move beyond simple co-activation analysis.

- The discovery of "MoE entanglement" (the long-range "stripes" in Fig 3d and 5a) is a major finding. It compellingly argues that MoE routing is a complex, non-local phenomenon, which has significant implications for how these models are understood.

**Weaknesses:**

Correlation vs. Causation: The analysis is entirely correlational. While it shows that the output of expert M1E9 correlates with routing decisions in layer 10, it does not prove a causal link. The paper lacks interventional experiments (e.g., ablating a high-variance expert and measuring the actual change in downstream routing) to confirm that these high-variance components are truly driving the decisions.

Underdeveloped Link to "Super Experts": The paper attempts to connect its high-influence experts to the "Super Experts" identified by Su et al. (2025) but finds the link is weak (Section 7, Appendix H). This is an interesting anti-finding, but it feels incomplete. It raises the question: which metric matters more? This paper's routing "influence" (variance) or the "Super Expert" output magnitude? The paper opens this door but doesn't walk through it.

Some of the notation are made deliberately complicate, which is hard to read at first, especially in section 3 and 4.1

**Questions:**

Could you elaborate on the `LN_bar`? from Eq 8, it seems that The methodology's decomposition (Eq. 8) relies on attributing scores to normalized components (`LN_bar`). However, Layer Normalization (LN) is a non-linear function applied to the sum of all components. $LN(x+y) ≠ LN(x) + LN(y)$. The paper defines $LN_bar(z) = \frac{(z * γ)}{RMS(total input)}$, which means the "contribution" of one component is scaled by a denominator that depends on all other components. So is such decomposition really make sense?

On Model Design Implications: Your findings suggest some layers/experts are far more important to routing. Based on this, would you recommend alternative MoE architectures? For instance, any new architecture designs or distribution of MoE layers in the network?

On Training Dynamics: This analysis is a snapshot of pre-trained models. Have you investigated how these influence maps evolve during training? Does the long-range "entanglement" (the stripes) emerge early, or is it a late-stage phenomenon as the model refines its pathways?

---

> ### Author Response · Authors · 2025-11-24
> **Official Comment by Authors (Part 1)**
>
> We thank you for your time and valuable feedback. We address the raised weaknesses and questions below.
>
> ## Response to the weaknesses:
>
> ### W1: Lack of Interventional Experiments
>
> **(Copied from General Response 3)** We have now added a case study of causal analysis in Section 7.2 and Appendix I. We show in a physical perspective that the expert M1E9 with the largest variance also has the highest influence on the top-K expert selections in following layers, compared with other experts in M1. Furthermore, we identify that M1E9 controls the activation of M4E14 and they need to coexist to exert a prominent influence on the routing decisions in MoE Layer 5 and the following layers. The results from the causal study confirm that these experts assigning high-variance scores tend to have a strong influence on the routing decisions and validate the effectiveness of our methods.
>
> ### W2: Discussion on "Influence" and "Super Experts"
>
> **(Copied from General Response 4)** Super experts (Su et al.) are characterized by having unusually large output magnitude, which implies they can have a strong impact on the model’s final output. This is an observation about representational strength at the level of model outputs.
> In contrast, our metric is designed to characterize cross-layer (both short- and long-range) correlations by quantifying how each component contributes to the expert score landscape. Rather than focusing on output dominance, our method serves a different purpose: it enables fine-grained mechanistic analysis of how routing signals are shaped and propagated across layers.
> In relation to the above, we also would like to emphasize that the result in Figure 5 (and line 472) in the paper demonstrate that not all the Super Experts have typically large scoring variance, i.e. are not necessarily contributing highly to all following layers. This indicates that the set of Super Experts and the large-variance experts found by our method are different although they may have an overlap.
>
> ### W3: Overcomplicated Notation
>
> We thank you for pointing this out. We will simplify the notations in the revised version to improve readability.
>
> ## Response to the questions:
>
> ### Q1: Does the proposed decomposition make sense?
>
> **(Copied from General Response 1)** Our decomposition uses $\mathrm{\overline{LN}}(c)= \frac{\gamma \odot c}{\mathrm{RMS}(c_{total})}$, where $c$ is a component vector and the denominator depends on the full input vector $c_{total}$. Therefore, the decomposition distributes the **same normalization factor** across all components. This preserves the direction of each component's contribution and keeps their relative magnitudes faithful to what the router actually receives. The decomposition should be viewed as **"what each component looks like after the shared normalization factor has been applied,"** rather than **“the output of LayerNorm applied to that component in isolation.”**
>
> Furthermore, we have now added a case study of direct causal analysis in Section 7.2 to validate the effectiveness of the proposed decomposition method.
>
> **(Additional Response)** We understand the concern that the RMSNorm function is not linear and hence $\mathrm{LN}(x+y) \ne \mathrm{LN}(x)+\mathrm{LN}(y)$. This operation $\mathrm{\overline{LN}}(c)$ is _not_ assuming linearity of RMSNorm; it simply isolates the linear term that the router weights interact with during the dot product.

---

> ### Author Response · Authors · 2025-11-24
> **Official Comment by Authors (Part 2)**
>
> ### Q2: On model design implications
>
> The analysis in the paper does naturally suggest several promising design directions:
>
> a) **Non-uniform MoE placement**. Our variance maps show that a few MoE layers consistently dominate the routing landscape, while others have comparatively modest contribution. This suggests exploring heterogeneous depthwise allocation, where MoE capacity (number of layers, width, or top-k) is concentrated in high-influence regions, and reduced or even removed in consistently low-influence regions.
>
> b) **Tiered expert capacity**. At the expert level, we observe that only a small subset of experts maintains strong contributions across many layers. This motivates architectures with tiered experts—for example, a mix of “high-capacity” experts in high-influence layers and cheaper experts elsewhere, rather than a uniform design.
>
> c) **Cross-layer–aware routing heuristics**. Since routing decisions are not purely local, one could use our cross-layer contribution maps as a prior for inference-time scheduling: e.g., prefetching experts that are systematically promoted by influential upstream layers, or biasing training-time regularizers to stabilize or diversify those long-range pathways.
>
> We have now included the information above in Appendix O. In the current manuscript we stop at identifying these patterns and outlining such ideas at a high level, because a careful architectural study (with new models, training, and ablations) would substantially expand the scope. We have added text to make this distinction explicit: our method provides actionable signals for where heterogeneous MoE placement or tiered experts may be most effective, but fully developing and evaluating new architectures is left for follow-up work.
>
> ### Q3: On training dynamics
>
> That indeed would be interesting to explore (thanks for the pointer). Due to the lack of time in the rebuttal stage for doing all required extra experiments, we could not include more results on that front. That being said, should the paper be accepted we commit to conducting experiment on released checkpoints of OLMoE.

---

> ### Author Response · Authors · 2025-11-28
>
> Gentle reminder: we would be more than happy to answer any question, or clarify any point.

---

### Author Response · Authors · 2025-11-24
**General response to all reviewers**

We thank all reviewers for their insightful feedback. During the discussion period, we received one comment from Reviewer NVZU, who confirmed that our clarifications addressed their concerns. Below, we address the five major themes raised across different reviews. We may also provide additional responses to the reviewer who raised the following questions.

**1. Validity of the proposed decomposition (Eq. 8)**

Our decomposition uses $\mathrm{\overline{LN}}(c)= \frac{\gamma \odot c}{\mathrm{RMS}(c_{total})}$, where $c$ is a component vector and the denominator depends on the full input vector $c_{total}$. Therefore, the decomposition distributes the **same normalization factor** across all components. This preserves the direction of each component's contribution and keeps their relative magnitudes faithful to what the router actually receives. The decomposition should be viewed as **"what each component looks like after the shared normalization factor has been applied,"** rather than **"the output of LayerNorm applied to that component in isolation."**

Furthermore, we have now added a case study of direct causal analysis in Section 7.2 to validate the effectiveness of the proposed decomposition method.

**2. Validity of variance as a metric for influence**

A routing decision is determined by the scores of all experts. Hence, we apply the variance of scores assigned by a component to all the experts in an MoE layer to measure how the component influences the routing decisions. The intuition is that a higher variance leads to a more uneven distribution of assignment scores, hence the corresponding component has a higher influence on the routing decisions. We have now added experiments to demonstrate that variance of the assignment score of a component can reflect its impact on the ranking of top-K experts (Section 7.2 and Appendix N).

**3. Causal Analysis**

We have now added a case study of causal analysis in Section 7.2 and Appendix I. We show in a physical perspective that the expert M1E9 with the largest variance also has the highest influence on the top-K expert selections in following layers, compared with other experts in M1. Furthermore, we identify that M1E9 controls the activation of M4E14 and they need to coexist to exert a prominent influence on the routing decisions in MoE Layer 5 and the following layers. The results from the causal study confirm that these experts assigning high-variance scores tend to have a strong influence on the routing decisions and validate the effectiveness of our methods.

**4. Comparison of Super Experts and high-variance experts**

Super experts (Su et al.) are characterized by having unusually large output magnitude, which implies they can have a strong impact on the model’s final output. This is an observation about representational strength at the level of model outputs.
In contrast, our metric is designed to characterize cross-layer (both short- and long-range) correlations by quantifying how each component contributes to the expert score landscape. Rather than focusing on output dominance, our method serves a different purpose: it enables fine-grained mechanistic analysis of how routing signals are shaped and propagated across layers.
In relation to the above, we also would like to emphasize that the result in Figure 5 (and line 472) in the paper demonstrate that not all the Super Experts have typically large scoring variance, i.e. are not necessarily contributing highly to all following layers. This indicates that the set of Super Experts and the large-variance experts found by our method are different although they may have an overlap.

**5. Generalizability of observed results**

We have now added experimental results on a math reasoning task dataset (Appendix J), which reveals a similar pattern of scoring (e.g., stripes and high-influence experts) that occurs in the C4 dataset.

**Finally, we apologize for the mistake of omitting the mandatory section on "THE USE OF LARGE LANGUAGE MODELS" in our first submission. We have strictly adhered to ICLR guidelines and have added the required "The Use of Large Language Models" section in Section 9.**

---

> ### Author Response · Authors · 2025-12-04
> **Revisions of the Manuscript**
>
> Revisions of the manuscript are highlighted in blue.
>
> The changes related to the reviewers' concerns are as follows:
> 1. (**Section 4.1**) We further clarified our decomposition method.
> 2. (**Section 5.1**) We further clarified the computation method of the metrics.
> 3. (**Section 7.2**) We added a causal analysis of the high-variance expert M1E9 in OLMoE.
> 4. (**Section 9**) We added a statement regarding LLM usage.
> 5. (**Appendix I~N**) We added the experimental results suggested by the reviewers:
> - 5.1	(**Appendix I**) We added a zero-ablation study on three experts in OLMoE.
> - 5.2	(**Appendix J**) We added an experiment on a math reasoning task to examine the generalizability of the findings.
> - 5.3	(**Appendix K**) We added an L2-norm heatmap for comparison with the variance of scores assigned by attention/MoE layers.
> - 5.4	(**Appendix L**) We added an analysis of sensitivity to the hyperparameter K (of the Top-K).
> - 5.5	(**Appendix M**) We added an experiment on swapping rank K experts with unselected experts to observe its influence.
> - 5.6	(**Appendix N**) We added the result of the Spearman coefficients.
> 6. (**Appendix O**) We added the model design implications.
> 7. (**Appendix P**) We added a brief mechanistic explanation of the observed phenomena.

---

### Meta-Review · Area_Chair_PXML · 2025-12-04

**Summary:**

Across the four reviews (Dj7p, tjsZ, T74p, NVZU), several consistent weaknesses emerged. Reviewers questioned the validity of the Eq. 8 decomposition (Dj7p, T74p), noting the nonlinearity of RMSNorm and potential attribution bias. Multiple reviewers (Dj7p, tjsZ, NVZU) raised concerns about using variance as the primary influence metric, especially its insensitivity to Top-K boundary effects. Others emphasized that the analysis was initially correlational and lacked causal evidence (Dj7p), and noted limitations in generality due to restricted model/task coverage (tjsZ, NVZU). Reviewers also pointed out notation and clarity issues and requested clearer mechanistic explanations (T74p).

After having read the reviews, rebuttals, and briefly the paper, given the novelty, I am aligned with the overall scores of the reviewers and recommend accepting the paper as most concerns that I have been raised been mostly addressed.

**Reviewer Concerns:**

see above

**Reviewer Scores:**

see above

---

### Decision · Program_Chairs · 2026-01-26

Accept (Poster)